# Streaming Sparse Gaussian Process Approximations

**Thang D. Bui**[*]      **Cuong V. Nguyen**[*]      **Richard E. Turner**
Department of Engineering, University of Cambridge, UK
{tdb40,vcn22,ret26}@cam.ac.uk

## Abstract

Sparse pseudo-point approximations for Gaussian process (GP) models provide a suite of methods that support deployment of GPs in the large data regime and enable analytic intractabilities to be sidestepped. However, the field lacks a principled method to handle streaming data in which both the posterior distribution over function values and the hyperparameter estimates are updated in an online fashion. The small number of existing approaches either use suboptimal hand-crafted heuristics for hyperparameter learning, or suffer from catastrophic forgetting or slow updating when new data arrive. This paper develops a new principled framework for deploying Gaussian process probabilistic models in the streaming setting, providing methods for learning hyperparameters and optimising pseudo-input locations. The proposed framework is assessed using synthetic and real-world datasets.

## 1 Introduction

Probabilistic models employing Gaussian processes have become a standard approach to solving many machine learning tasks, thanks largely to the modelling flexibility, robustness to overfitting, and well-calibrated uncertainty estimates afforded by the approach [1]. One of the pillars of the modern Gaussian process probabilistic modelling approach is a set of sparse approximation schemes that allow the prohibitive computational cost of GP methods, typically $\mathcal{O}(N^3)$ for training and $\mathcal{O}(N^2)$ for prediction where $N$ is the number of training points, to be substantially reduced whilst still retaining accuracy. Arguably the most important and influential approximations of this sort are pseudo-point approximation schemes that employ a set of $M \ll N$ pseudo-points to summarise the observational data thereby reducing computational costs to $\mathcal{O}(NM^2)$ and $\mathcal{O}(M^2)$ for training and prediction, respectively [2, 3]. Stochastic optimisation methods that employ mini-batches of training data can be used to further reduce computational costs [4, 5, 6, 7], allowing GPs to be scaled to datasets comprising millions of data points.

The focus of this paper is to provide a comprehensive framework for deploying the Gaussian process probabilistic modelling approach to streaming data. That is, data that arrive sequentially in an online fashion, possibly in small batches, and whose number are not known a priori (and indeed may be infinite). The vast majority of previous work has focussed exclusively on the batch setting and there is not a satisfactory framework that supports learning and approximation in the streaming setting. A naïve approach might simply incorporate each new datum as they arrived into an ever-growing dataset and retrain the GP model from scratch each time. With infinite computational resources, this approach is optimal, but in the majority of practical settings, it is intractable. A feasible alternative would train on just the most recent $K$ training data points, but this completely ignores potentially large amounts of informative training data and it does not provide a method for incorporating the old model into the new one which would save computation (except perhaps through initialisation of the hyperparameters). Existing, sparse approximation schemes could be applied in the same manner, but they merely allow $K$ to be increased, rather than allowing all previous data to be leveraged, and again do not utilise intermediate approximate fits.

---

[*]These authors contributed equally to this work.

What is needed is a method for performing learning and sparse approximation that incrementally updates the previously fit model using the new data. Such an approach would utilise all the previous training data (as they will have been incorporated into the previously fit model) and leverage as much of the previous computation as possible at each stage (since the algorithm only requires access to the data at the current time point). Existing *stochastic* sparse approximation methods could potentially be used by collecting the streamed data into mini-batches. However, the assumptions underpinning these methods are ill-suited to the streaming setting and they perform poorly (see sections 2 and 4).

This paper provides a new principled framework for deploying Gaussian process probabilistic models in the streaming setting. The framework subsumes Csató and Opper's two seminal approaches to online regression [8, 9] that were based upon the variational free energy (VFE) and expectation propagation (EP) approaches to approximate inference respectively. In the new framework, these algorithms are recovered as special cases. We also provide principled methods for learning hyperparameters (learning was not treated in the original work and the extension is non-trivial) and optimising pseudo-input locations (previously handled via hand-crafted heuristics). The approach also relates to the streaming variational Bayes framework [10]. We review background material in the next section and detail the technical contribution in section 3, followed by several experiments on synthetic and real-world data in section 4.

## 2    Background

Regression models that employ Gaussian processes are state of the art for many datasets [11]. In this paper we focus on the simplest GP regression model as a test case of the streaming framework for inference and learning. Given $N$ input and real-valued output pairs $\{\mathbf{x}_n, y_n\}_{n=1}^N$, a standard GP regression model assumes $y_n = f(\mathbf{x}_n) + \epsilon_n$, where $f$ is an unknown function that is corrupted by Gaussian observation noise $\epsilon_n \sim \mathcal{N}(0, \sigma_y^2)$. Typically, $f$ is assumed to be drawn from a zero-mean GP prior $f \sim \mathcal{GP}(\mathbf{0}, k(\cdot, \cdot|\theta))$ whose covariance function depends on hyperparameters $\theta$. In this simple model, the posterior over $f$, $p(f|\mathbf{y}, \theta)$, and the marginal likelihood $p(\mathbf{y}|\theta)$ can be computed analytically (here we have collected the observations into a vector $\mathbf{y} = \{y_n\}_{n=1}^N$).[2] However, these quantities present a computational challenge resulting in an $O(N^3)$ complexity for maximum likelihood training and $O(N^2)$ per test point for prediction.

This prohibitive complexity of exact learning and inference in GP models has driven the development of many sparse approximation frameworks [12, 13]. In this paper, we focus on the variational free energy approximation scheme [3, 14] which lower bounds the marginal likelihood of the data using a variational distribution $q(f)$ over the latent function:

$$\log p(\mathbf{y}|\theta) = \log \int \mathrm{d}f\; p(\mathbf{y}, f|\theta) \geq \int \mathrm{d}f\; q(f) \log \frac{p(\mathbf{y}, f|\theta)}{q(f)} = \mathcal{F}_{\mathrm{vfe}}(q, \theta). \qquad (1)$$

Since $\mathcal{F}_{\mathrm{vfe}}(q, \theta) = \log p(\mathbf{y}|\theta) - \mathrm{KL}[q(f)||p(f|\mathbf{y}, \theta)]$, where $\mathrm{KL}[\cdot||\cdot]$ denotes the Kullback–Leibler divergence, maximising this lower bound with respect to $q(f)$ guarantees the approximate posterior gets *closer* to the exact posterior $p(f|\mathbf{y}, \theta)$. Moreover, the variational bound $\mathcal{F}_{\mathrm{vfe}}(q, \theta)$ approximates the marginal likelihood and can be used for learning the hyperparameters $\theta$.

In order to arrive at a computationally tractable method, the approximate posterior is parameterized via a set of $M$ pseudo-points $\mathbf{u}$ that are a subset of the function values $f = \{f_{\neq \mathbf{u}}, \mathbf{u}\}$ and which will summarise the data. Specifically, the approximate posterior is assumed to be $q(f) = p(f_{\neq \mathbf{u}}|\mathbf{u}, \theta)q(\mathbf{u})$, where $q(\mathbf{u})$ is a variational distribution over $\mathbf{u}$ and $p(f_{\neq \mathbf{u}}|\mathbf{u}, \theta)$ is the prior distribution of the remaining latent function values. This assumption allows the following critical cancellation that results in a computationally tractable lower bound:

$$\mathcal{F}_{\mathrm{vfe}}(q(\mathbf{u}), \theta) = \int \mathrm{d}f\; q(f) \log \frac{p(\mathbf{y}|f, \theta)p(\mathbf{u}|\theta)\widecancel{p(f_{\neq \mathbf{u}}|\mathbf{u}, \theta)}}{\widecancel{p(f_{\neq \mathbf{u}}|\mathbf{u}, \theta)}q(\mathbf{u})}$$

$$= -\mathrm{KL}[q(\mathbf{u})||p(\mathbf{u}|\theta)] + \sum_n \int \mathrm{d}\mathbf{u}\; q(\mathbf{u})p(f_n|\mathbf{u}, \theta) \log p(y_n|f_n, \theta),$$

where $f_n = f(\mathbf{x}_n)$ is the latent function value at $\mathbf{x}_n$. For the simple GP regression model considered here, closed-form expressions for the optimal variational approximation $q_{\mathrm{vfe}}(f)$ and the optimal

variational bound $\mathcal{F}_{\text{vfe}}(\theta) = \max_{q(\mathbf{u})} \mathcal{F}_{\text{vfe}}(q(\mathbf{u}), \theta)$ (also called the 'collapsed' bound) are available:

$$p(f|\mathbf{y}, \theta) \approx q_{\text{vfe}}(f) \propto p(f_{\neq \mathbf{u}}|\mathbf{u}, \theta)p(\mathbf{u}|\theta)\mathcal{N}(\mathbf{y}; \mathbf{K_{fu}K_{uu}^{-1}u}, \sigma_y^2\mathbf{I}),$$

$$\log p(\mathbf{y}|\theta) \approx \mathcal{F}_{\text{vfe}}(\theta) = \log \mathcal{N}(\mathbf{y}; \mathbf{0}, \mathbf{K_{fu}K_{uu}^{-1}K_{uf}} + \sigma_y^2\mathbf{I}) - \frac{1}{2\sigma_y^2}\sum_n (k_{nn} - \mathbf{K}_{n\mathbf{u}}\mathbf{K_{uu}^{-1}}\mathbf{K}_{\mathbf{u}n}),$$

where $\mathbf{f}$ is the latent function values at training points, and $\mathbf{K_{f_1 f_2}}$ is the covariance matrix between the latent function values $\mathbf{f}_1$ and $\mathbf{f}_2$. Critically, the approach leads to $O(NM^2)$ complexity for approximate maximum likelihood learning and $O(M^2)$ per test point for prediction. In order for this method to perform well, it is necessary to adapt the pseudo-point input locations, e.g. by optimising the variational free energy, so that the pseudo-data distribute themselves over the training data.

Alternatively, stochastic optimisation may be applied directly to the original, *uncollapsed* version of the bound [4, 15]. In particular, an unbiased estimate of the variational lower bound can be obtained using a small number of training points randomly drawn from the training set:

$$\mathcal{F}_{\text{vfe}}(q(\mathbf{u}), \theta) \approx -\text{KL}[q(\mathbf{u})||p(\mathbf{u}|\theta)] + \frac{N}{|B|}\sum_{y_n \in B}\int d\mathbf{u}\, q(\mathbf{u})p(f_n|\mathbf{u}, \theta)\log p(y_n|f_n, \theta).$$

Since the optimal approximation is Gaussian as shown above, $q(\mathbf{u})$ is often posited as a Gaussian distribution and its parameters are updated by following the (noisy) gradients of the stochastic estimate of the variational lower bound. By passing through the training set a sufficient number of times, the variational distribution converges to the optimal solution above, given appropriately decaying learning rates [4].

In principle, the stochastic uncollapsed approach is applicable to the streaming setting as it refines an approximate posterior based on mini-batches of data that can be considered to arrive sequentially (here $N$ would be the number of data points seen so far). However, it is unsuited to this task since stochastic optimisation assumes that the data subsampling process is *uniformly random*, that the training set is revisited multiple times, and it typically makes a single gradient update per mini-batch. These assumptions are incompatible with the streaming setting: continuously arriving data are not typically drawn iid from the input distribution (consider an evolving time-series, for example); the data can only be touched once by the algorithm and not revisited due to computational constraints; each mini-batch needs to be processed intensively as it will not be revisited (multiple gradient steps would normally be required, for example, and this runs the risk of forgetting old data without delicately tuning the learning rates). In the following sections, we shall discuss how to tackle these challenges through a novel online inference and learning procedure, and demonstrate the efficacy of this method over the uncollapsed approach and naïve online versions of the collapsed approach.

## 3 Streaming sparse GP (SSGP) approximation using variational inference

The general situation assumed in this paper is that data arrive sequentially so that at each step new data points $\mathbf{y}_{\text{new}}$ are added to the old dataset $\mathbf{y}_{\text{old}}$. The goal is to approximate the marginal likelihood and the posterior of the latent process at each step, which can be used for anytime prediction. The hyperparameters will also be adjusted online. Importantly, we assume that we can only access the current data points $\mathbf{y}_{\text{new}}$ directly for computational reasons (it might be too expensive to hold $\mathbf{y}_{\text{old}}$ and $\mathbf{x}_{1:N_{\text{old}}}$ in memory, for example, or approximations made at the previous step must be reused to reduce computational overhead). So the effect of the old data on the current posterior must be propagated through the previous posterior. We will now develop a new sparse variational free energy approximation for this purpose, that compactly summarises the old data via pseudo-points. The pseudo-inputs will also be adjusted online since this is critical as new parts of the input space will be revealed over time. The framework is easily extensible to more complex non-linear models.

### 3.1 Online variational free energy inference and learning

Consider an approximation to the true posterior at the previous step, $q_{\text{old}}(f)$, which must be updated to form the new approximation $q_{\text{new}}(f)$,

$$q_{\text{old}}(f) \approx p(f|\mathbf{y}_{\text{old}}) = \frac{1}{\mathcal{Z}_1(\theta_{\text{old}})}p(f|\theta_{\text{old}})p(\mathbf{y}_{\text{old}}|f), \tag{2}$$

$$q_{\text{new}}(f) \approx p(f|\mathbf{y}_{\text{old}}, \mathbf{y}_{\text{new}}) = \frac{1}{\mathcal{Z}_2(\theta_{\text{new}})}p(f|\theta_{\text{new}})p(\mathbf{y}_{\text{old}}|f)p(\mathbf{y}_{\text{new}}|f). \tag{3}$$

Whilst the updated exact posterior $p(f|\mathbf{y}_{\text{old}}, \mathbf{y}_{\text{new}})$ balances the contribution of old and new data through their likelihoods, the new approximation cannot access $p(\mathbf{y}_{\text{old}}|f)$ directly. Instead, we can find an approximation of $p(\mathbf{y}_{\text{old}}|f)$ by inverting eq. (2), that is $p(\mathbf{y}_{\text{old}}|f) \approx \mathcal{Z}_1(\theta_{\text{old}})q_{\text{old}}(f)/p(f|\theta_{\text{old}})$. Substituting this into eq. (3) yields,

$$\hat{p}(f|\mathbf{y}_{\text{old}}, \mathbf{y}_{\text{new}}) = \frac{\mathcal{Z}_1(\theta_{\text{old}})}{\mathcal{Z}_2(\theta_{\text{new}})} p(f|\theta_{\text{new}}) p(\mathbf{y}_{\text{new}}|f) \frac{q_{\text{old}}(f)}{p(f|\theta_{\text{old}})}. \tag{4}$$

Although it is tempting to use this as the new posterior, $q_{\text{new}}(f) = \hat{p}(f|\mathbf{y}_{\text{old}}, \mathbf{y}_{\text{new}})$, this recovers exact GP regression with fixed hyperparameters (see section 3.3) and it is intractable. So, instead, we consider a variational update that projects the distribution back to a tractable form using pseudo-data. At this stage we allow the pseudo-data input locations in the new approximation to differ from those in the old one. This is required if new regions of input space are gradually revealed, as for example in typical time-series applications. Let $\mathbf{a} = f(\mathbf{z}_{\text{old}})$ and $\mathbf{b} = f(\mathbf{z}_{\text{new}})$ be the function values at the pseudo-inputs before and after seeing new data. Note that the number of pseudo-points, $M_{\mathbf{a}} = |\mathbf{a}|$ and $M_{\mathbf{b}} = |\mathbf{b}|$ are not necessarily restricted to be the same. The form of the approximate posterior mirrors that in the batch case, that is, the previous approximate posterior, $q_{\text{old}}(f) = p(f_{\neq \mathbf{a}}|\mathbf{a}, \theta_{\text{old}})q_{\text{old}}(\mathbf{a})$ where we assume $q_{\text{old}}(\mathbf{a}) = \mathcal{N}(\mathbf{a}; \mathbf{m}_{\mathbf{a}}, \mathbf{S}_{\mathbf{a}})$. The new posterior approximation takes the same form, but with the new pseudo-points and new hyperparameters: $q_{\text{new}}(f) = p(f_{\neq \mathbf{b}}|\mathbf{b}, \theta_{\text{new}})q_{\text{new}}(\mathbf{b})$. Similar to the batch case, this approximate inference problem can be turned into an optimisation problem using variational inference. Specifically, consider

$$\text{KL}[q_{\text{new}}(f)||\hat{p}(f|\mathbf{y}_{\text{old}}, \mathbf{y}_{\text{new}})] = \int df \, q_{\text{new}}(f) \log \frac{p(f_{\neq \mathbf{b}}|\mathbf{b}, \theta_{\text{new}})q_{\text{new}}(\mathbf{b})}{\frac{\mathcal{Z}_1(\theta_{\text{old}})}{\mathcal{Z}_2(\theta_{\text{new}})}p(f|\theta_{\text{new}})p(\mathbf{y}_{\text{new}}|f)\frac{q_{\text{old}}(f)}{p(f|\theta_{\text{old}})}} \tag{5}$$

$$= \log \frac{\mathcal{Z}_2(\theta_{\text{new}})}{\mathcal{Z}_1(\theta_{\text{old}})} + \int df \, q_{\text{new}}(f) \left[ \log \frac{p(\mathbf{a}|\theta_{\text{old}})q_{\text{new}}(\mathbf{b})}{p(\mathbf{b}|\theta_{\text{new}})q_{\text{old}}(\mathbf{a})p(\mathbf{y}_{\text{new}}|f)} \right].$$

Since the KL divergence is non-negative, the second term in the expression above is the negative approximate lower bound of the online log marginal likelihood (as $\mathcal{Z}_2/\mathcal{Z}_1 \approx p(\mathbf{y}_{\text{new}}|\mathbf{y}_{\text{old}})$), or the variational free energy $\mathcal{F}(q_{\text{new}}(f), \theta_{\text{new}})$. By setting the derivative of $\mathcal{F}$ w.r.t. $q(\mathbf{b})$ equal to 0, the optimal approximate posterior can be obtained for the regression case,[3]

$$q_{\text{vfe}}(\mathbf{b}) \propto p(\mathbf{b}) \exp \left( \int d\mathbf{a} \, p(\mathbf{a}|\mathbf{b}) \log \frac{q_{\text{old}}(\mathbf{a})}{p(\mathbf{a}|\theta_{\text{old}})} + \int d\mathbf{f} \, p(\mathbf{f}|\mathbf{b}) \log p(\mathbf{y}_{\text{new}}|\mathbf{f}) \right) \tag{6}$$

$$\propto p(\mathbf{b})\mathcal{N}(\hat{\mathbf{y}}; \mathbf{K}_{\hat{\mathbf{f}}\mathbf{b}}\mathbf{K}_{\mathbf{bb}}^{-1}\mathbf{b}, \Sigma_{\hat{\mathbf{y}}, \text{vfe}}), \tag{7}$$

where $\mathbf{f}$ is the latent function values at the new training points,

$$\hat{\mathbf{y}} = \begin{bmatrix} \mathbf{y}_{\text{new}} \\ \mathbf{D}_{\mathbf{a}}\mathbf{S}_{\mathbf{a}}^{-1}\mathbf{m}_{\mathbf{a}} \end{bmatrix}, \quad \mathbf{K}_{\hat{\mathbf{f}}\mathbf{b}} = \begin{bmatrix} \mathbf{K}_{\mathbf{fb}} \\ \mathbf{K}_{\mathbf{ab}} \end{bmatrix}, \quad \Sigma_{\hat{\mathbf{y}}, \text{vfe}} = \begin{bmatrix} \sigma_y^2\mathbf{I} & \mathbf{0} \\ \mathbf{0} & \mathbf{D}_{\mathbf{a}} \end{bmatrix}, \quad \mathbf{D}_{\mathbf{a}} = (\mathbf{S}_{\mathbf{a}}^{-1} - \mathbf{K}_{\mathbf{aa}}'^{-1})^{-1}.$$

The negative variational free energy is also analytically available,

$$\mathcal{F}(\theta) = \log \mathcal{N}(\hat{\mathbf{y}}; \mathbf{0}, \mathbf{K}_{\hat{\mathbf{f}}\mathbf{b}}\mathbf{K}_{\mathbf{bb}}^{-1}\mathbf{K}_{\mathbf{b}\hat{\mathbf{f}}} + \Sigma_{\hat{\mathbf{y}}, \text{vfe}}) - \frac{1}{2\sigma_y^2}\text{tr}(\mathbf{K}_{\mathbf{ff}} - \mathbf{K}_{\mathbf{fb}}\mathbf{K}_{\mathbf{bb}}^{-1}\mathbf{K}_{\mathbf{bf}}) + \Delta_{\mathbf{a}}; \text{ where} \tag{8}$$

$$2\Delta_{\mathbf{a}} = -\log|\mathbf{S}_{\mathbf{a}}| + \log|\mathbf{K}_{\mathbf{aa}}'| + \log|\mathbf{D}_{\mathbf{a}}| + \mathbf{m}_{\mathbf{a}}^{\mathsf{T}}(\mathbf{S}_{\mathbf{a}}^{-1}\mathbf{D}_{\mathbf{a}}\mathbf{S}_{\mathbf{a}}^{-1} - \mathbf{S}_{\mathbf{a}}^{-1})\mathbf{m}_{\mathbf{a}} - \text{tr}[\mathbf{D}_{\mathbf{a}}^{-1}\mathbf{Q}_{\mathbf{a}}] + \text{const.}$$

Equations (7) and (8) provide the complete recipe for online posterior update and hyperparameter learning in the streaming setting. The computational complexity and memory overhead of the new method is of the same order as the uncollapsed stochastic variational inference approach. The procedure is demonstrated on a toy regression example as shown in fig. 1[Left].

## 3.2 Online $\alpha$-divergence inference and learning

One obvious extension of the online approach discussed above replaces the KL divergence in eq. (5) with a more general $\alpha$-divergence [16]. This does not affect tractability: the optimal form of the approximate posterior can be obtained analytically for the regression case, $q_{\text{pep}}(\mathbf{b}) \propto p(\mathbf{b})\mathcal{N}(\hat{\mathbf{y}}; \mathbf{K}_{\hat{\mathbf{f}}\mathbf{b}}\mathbf{K}_{\mathbf{bb}}^{-1}\mathbf{b}, \Sigma_{\hat{\mathbf{y}}, \text{pep}})$ where

$$\Sigma_{\hat{\mathbf{y}}, \text{pep}} = \begin{bmatrix} \sigma_y^2\mathbf{I} + \alpha\,\text{diag}(\mathbf{K}_{\mathbf{ff}} - \mathbf{K}_{\mathbf{fb}}\mathbf{K}_{\mathbf{bb}}^{-1}\mathbf{K}_{\mathbf{bf}}) & \mathbf{0} \\ \mathbf{0} & \mathbf{D}_{\mathbf{a}} + \alpha(\mathbf{K}_{\mathbf{aa}} - \mathbf{K}_{\mathbf{ab}}\mathbf{K}_{\mathbf{bb}}^{-1}\mathbf{K}_{\mathbf{ba}}) \end{bmatrix}. \tag{9}$$

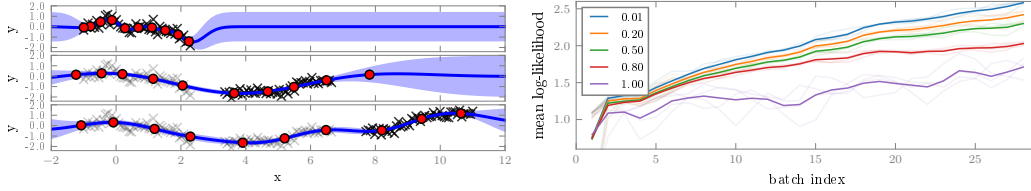

Figure 1: [Left] SSGP inference and learning on a toy time-series using the VFE approach. The black crosses are data points (past points are greyed out), the red circles are pseudo-points, and blue lines and shaded areas are the marginal predictive means and confidence intervals at test points. [Right] Log-likelihood of test data as training data arrives for different $\alpha$ values, for the pseudo periodic dataset (see section 4.2). We observed that $\alpha = 0.01$ is virtually identical to VFE. Dark lines are means over 4 splits and shaded lines are results for each split. Best viewed in colour.

This reduces back to the variational case as $\alpha \to 0$ (compare to eq. (7)) since then the $\alpha$-divergence is equivalent to the KL divergence. The approximate online log marginal likelihood is also analytically tractable and recovers the variational case when $\alpha \to 0$. Full details are provided in the appendix.

### 3.3   Connections to previous work and special cases

This section briefly highlights connections between the new framework and existing approaches including Power Expectation Propagation (Power-EP), Expectation Propagation (EP), Assumed Density Filtering (ADF), and streaming variational Bayes.

Recent work has unified a range of batch sparse GP approximations as special cases of the Power-EP algorithm [13]. The online $\alpha$-divergence approach to inference and learning described in the last section is equivalent to running a forward filtering pass of Power-EP. In other words, the current work generalizes the unifying framework to the streaming setting.

When the hyperparameters and the pseudo-inputs are fixed, $\alpha$-divergence inference for sparse GP regression recovers the batch solutions provided by Power-EP. In other words, only a single pass through the data is necessary for Power-EP to converge in sparse GP regression. For the case $\alpha = 1$, which is called Expectation Propagation, we recover the seminal work by Csató and Opper [8]. For the variational free energy case (equivalently where $\alpha \to 0$) we recover the seminal work by Csató [9]. The new framework can be seen to extend these methods to allow principled learning and pseudo-input optimisation. Interestingly, in the setting where hyperparameters and the pseudo-inputs are fixed, if pseudo-points are added at each stage at the new data input locations, the method returns the true posterior and marginal likelihood (see appendix).

For fixed hyperparameters and pseudo-points, the new VFE framework is equivalent to the application of streaming variational Bayes (VB) or online variational inference [10, 17, 18] to the GP setting in which the previous posterior plays a role of an effective prior for the new data. Similarly, the equivalent algorithm when $\alpha = 1$ is called Assumed Density Filtering [19]. When the hyperparameters are updated, the new method proposed here is different from streaming VB and standard application of ADF, as the new method propagates approximations to just the old likelihood terms and not the prior. Importantly, we found vanilla application of the streaming VB framework performed catastrophically for hyperparameter learning, so the modification is critical.

## 4   Experiments

In this section, the SSGP method is evaluated in terms of speed, memory usage, and accuracy (log-likelihood and error). The method was implemented on GPflow [20] and compared against GPflow's version of the following baselines: exact GP (GP), sparse GP using the collapsed bound (SGP), and stochastic variational inference using the uncollapsed bound (SVI). In all the experiments, the RBF kernel with ARD lengthscales is used, but this is not a limitation required by the new methods. An implementation of the proposed method can be found at http://github.com/thangbui/streaming_sparse_gp. Full experimental results and additional discussion points are included in the appendix.

### 4.1   Synthetic data

**Comparing $\alpha$-divergences**. We first consider the general online $\alpha$-divergence inference and learning framework and compare the performance of different $\alpha$ values on a toy online regression dataset

in fig. 1[Right]. Whilst the variational approach performs well, adapting pseudo-inputs to cover new regions of input space as they are revealed, algorithms using higher $\alpha$ values perform more poorly. Interestingly this appears to be related to the tendency for EP, in batch settings, to clump pseudo-inputs on top of one another [21]. Here the effect is much more extreme as the clumps accumulate over time, leading to a shortage of pseudo-points if the input range of the data increases. Although heuristics could be introduced to break up the clumps, this result suggests that using small $\alpha$ values for online inference and learning might be more appropriate (this recommendation differs from the batch setting where intermediate settings of $\alpha$ around 0.5 are best [13]). Due to these findings, for the rest of the paper, we focus on the variational case.

**Hyperparameter learning**. We generated multiple time-series from GPs with known hyperparameters and observation noises, and tracked the hyperparameters learnt by the proposed online variational free energy method and exact GP regression. Overall, SSGP can track and learn good hyperparameters, and if there are sufficient pseudo-points, it performs comparatively to full GP on the entire dataset. Interestingly, all models including full GP regression tend to learn bigger noise variances as any discrepancy in the true and learned function values is absorbed into this parameter.

### 4.2 Speed versus accuracy

In this experiment, we compare SSGP to the baselines (GP, SGP, and SVI) in terms of a speed-accuracy trade-off where the mean marginal log-likelihood (MLL) and the root mean squared error (RMSE) are plotted against the accumulated running time of each method after each iteration. The comparison is performed on two time-series datasets and a spatial dataset.

**Time-series data**. We first consider modelling a segment of the pseudo periodic synthetic dataset [22], previously used for testing indexing schemes in time-series databases. The segment contains 24,000 time-steps. Training and testing sets are chosen interleaved so that their sizes are both 12,000. The second dataset is an audio signal prediction dataset, produced from the TIMIT database [23] and previously used to evaluate GP approximations [24]. The signal was shifted down to the baseband and a segment of length 18,000 was used to produce interleaved training and testing sets containing 9,000 time steps. For both datasets, we linearly scale the input time steps to the range $[0, 10]$.

All algorithms are assessed in the mini-batch streaming setting with data $\mathbf{y}_{\text{new}}$ arriving in batches of size 300 and 500 taken in order from the time-series. The first 1,000 examples are used as an initial training set to obtain a reasonable starting model for each algorithm. In this experiment, we use memory-limited versions of GP and SGP that store the last 3,000 examples. This number was chosen so that the running times of these algorithms match those of SSGP or are slightly higher. For all sparse methods (SSGP, SGP, and SVI), we run the experiments with 100 and 200 pseudo-points.

For SVI, we allow the algorithm to make 100 stochastic gradient updates during each iteration and run preliminary experiments to compare 3 learning rates $r = 0.001, 0.01$, and $0.1$. The preliminary results showed that the performance of SVI was not significantly altered and so we only present the results for $r = 0.1$.

Figure 2 shows the plots of the accumulated running time (total training and testing time up until the current iteration) against the MLL and RMSE for the considered algorithms. It is clear that SSGP significantly outperforms the other methods both in terms of the MLL and RMSE, once sufficient training data have arrived. The performance of SSGP improves when the number of pseudo-points increases, but the algorithm runs more slowly. In contrast, the performance of GP and SGP, even after seeing more data or using more pseudo-points, does not increase significantly since they can only model a limited amount of data (the last 3,000 examples).

**Spatial data**. The second set of experiments consider the OS Terrain 50 dataset that contains spot heights of landscapes in Great Britain computed on a grid.[4] A block of $200 \times 200$ points was split into 10,000 training examples and 30,000 interleaved testing examples. Mini-batches of data of size 750 and 1,000 arrive in spatial order. The first 1,000 examples were used as an initial training set. For this dataset, we allow GP and SGP to remember the last 7,500 examples and use 400 and 600 pseudo-points for the sparse models. Figure 3 shows the results for this dataset. SSGP performs better than the other baselines in terms of the RMSE although it is worse than GP and SGP in terms of the MLL.

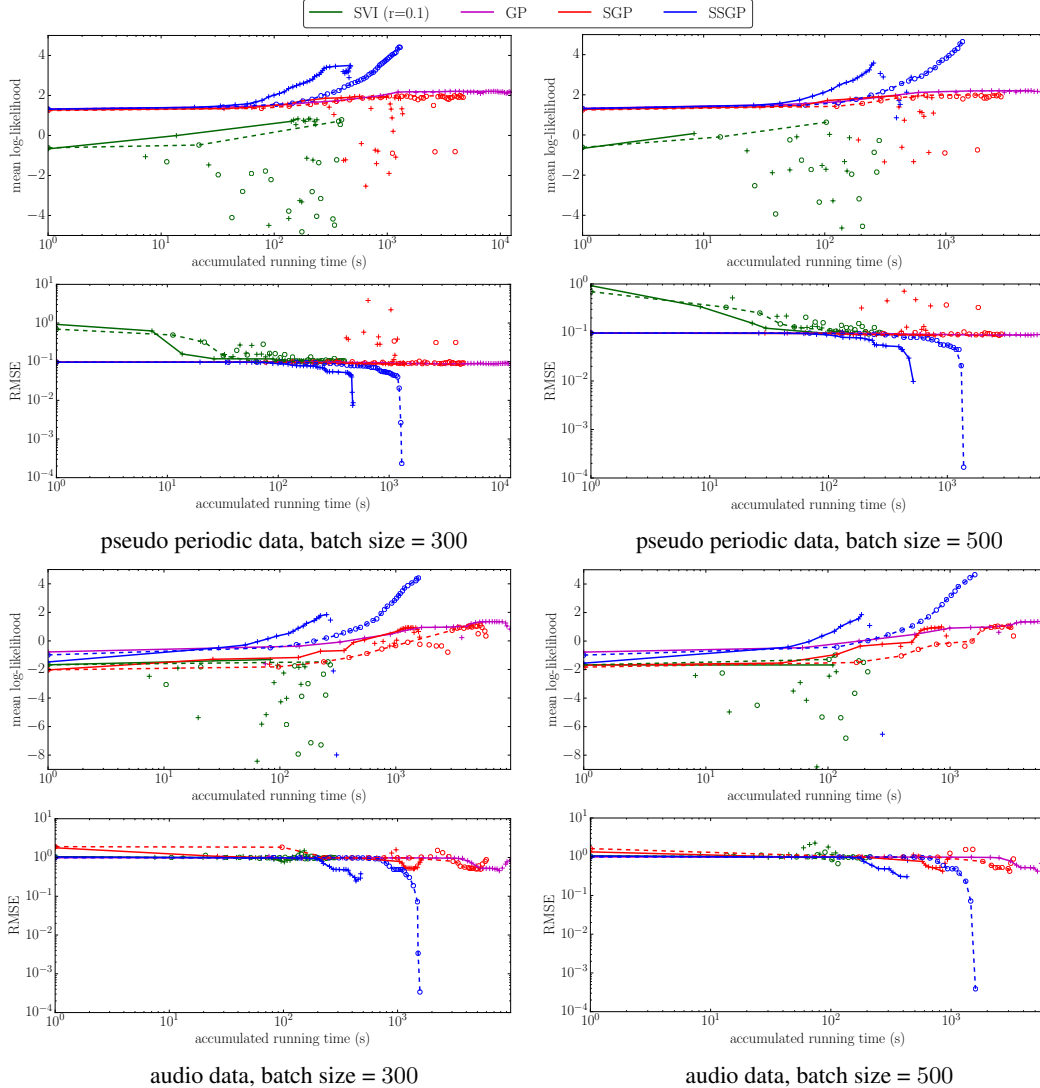

Figure 2: Results for time-series datasets with batch sizes 300 and 500. Pluses and circles indicate the results for $M = 100, 200$ pseudo-points respectively. For each algorithm (except for GP), the solid and dashed lines are the efficient frontier curves for $M = 100, 200$ respectively.

### 4.3 Memory usage versus accuracy

Besides running time, memory usage is another important factor that should be considered. In this experiment, we compare the memory usage of SSGP against GP and SGP on the Terrain dataset above with batch size 750 and $M = 600$ pseudo-points. We allow GP and SGP to use the last 2,000 and 6,000 examples for training, respectively. These numbers were chosen so that the memory usage of the two baselines roughly matches that of SSGP. Figure 4 plots the maximum memory usage of the three methods against the MLL and RMSE. From the figure, SSGP requires small memory usage while it can achieve comparable or better MLL and RMSE than GP and SGP.

### 4.4 Binary classification

We show a preliminary result for GP models with non-Gaussian likelihoods, in particular, a binary classification model on the benchmark *banana* dataset. As the optimal form for the approximate posterior is not analytically tractable, the uncollapsed variational free energy is optimised numerically. The predictions made by SSGP in a non-iid streaming setting are shown in fig. 5. SSGP performs well and achieves the performance of the batch sparse variational method [5].

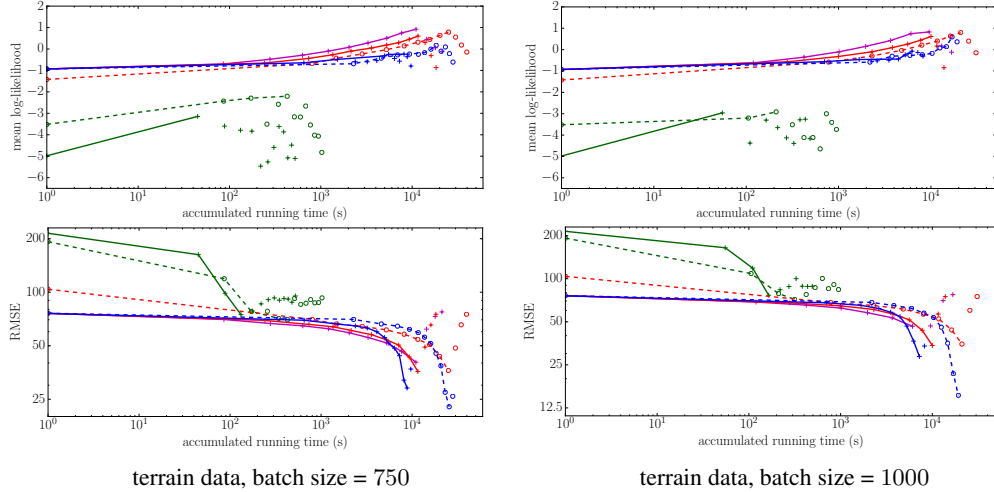

terrain data, batch size = 750          terrain data, batch size = 1000

Figure 3: Results for spatial data (see fig. 2 for the legend). Pluses/solid lines and circles/dashed lines indicate the results for $M = 400, 600$ pseudo-points respectively.

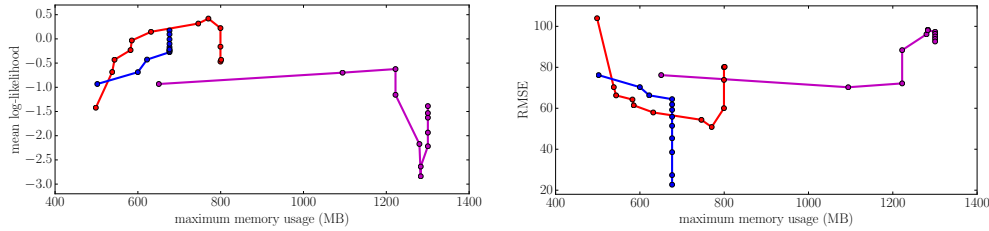

Figure 4: Memory usage of SSGP (blue), GP (magenta) and SGP (red) against MLL and RMSE.

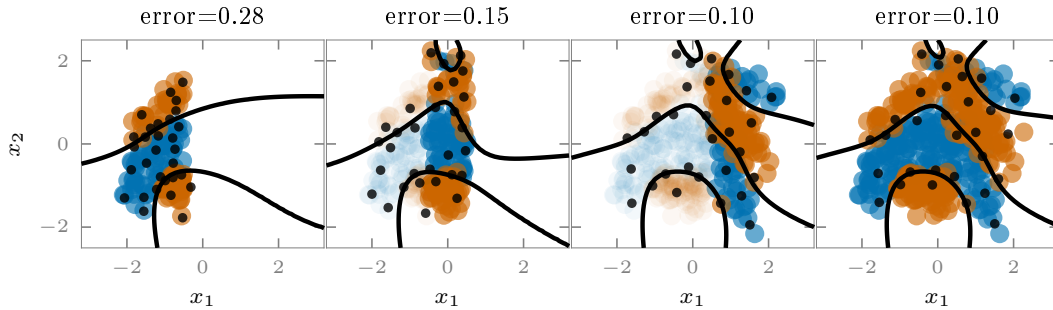

Figure 5: SSGP inference and learning on a binary classification task in a non-iid streaming setting. The right-most plot shows the prediction made by using sparse variational inference on full training data [5] for comparison. Past observations are greyed out. The pseudo-points are shown as black dots and the black curves show the decision boundary.

## 5 Summary

We have introduced a novel online inference and learning framework for Gaussian process models. The framework unifies disparate methods in the literature and greatly extends them, allowing sequential updates of the approximate posterior and online hyperparameter optimisation in a principled manner. The proposed approach outperforms existing approaches on a wide range of regression datasets and shows promising results on a binary classification dataset. A more thorough investigation on models with non-Gaussian likelihoods is left as future work. We believe that this framework will be particularly useful for efficient deployment of GPs in sequential decision making problems such as active learning, Bayesian optimisation, and reinforcement learning.

**Acknowledgements**

The authors would like to thank Mark Rowland, John Bradshaw, and Yingzhen Li for insightful comments and discussion. Thang D. Bui is supported by the Google European Doctoral Fellowship. Cuong V. Nguyen is supported by EPSRC grant EP/M0269571. Richard E. Turner is supported by Google as well as EPSRC grants EP/M0269571 and EP/L000776/1.

## Footnotes

[2]The dependence on the inputs $\{\mathbf{x}_n\}_{n=1}^N$ of the posterior, marginal likelihood, and other quantities is suppressed throughout to lighten the notation.

[3]Note that we have dropped $\theta_{\text{new}}$ from $p(\mathbf{b}|\theta_{\text{new}})$, $p(\mathbf{a}|\mathbf{b}, \theta_{\text{new}})$ and $p(\mathbf{f}|\mathbf{b}, \theta_{\text{new}})$ to lighten the notation.

[4]The dataset is available at: `https://data.gov.uk/dataset/os-terrain-50-dtm`.

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
