[Supplementary Material]

# Streaming Sparse Gaussian Process Approximations: Appendix

**Thang D. Bui**[*]  **Cuong V. Nguyen**[*]  **Richard E. Turner**
Department of Engineering, University of Cambridge, UK
{tdb40,vcn22,ret26}@cam.ac.uk

## 1 More discussions on the paper

### 1.1 Can the variational lower bound be derived using Jensen's inequality?

Yes. There are two equivalent ways of deriving VI:

1. Applying Jensen's inequality directly to the log marginal likelihood.
2. Explicitly writing down the $\mathrm{KL}(q\|p)$, noting that it is non-negative and rearranging to get the same bound as in (1).

(1) is often used in traditional VI literature. Many recent papers (e.g. [1] and our paper) use (2).

### 1.2 Comparison to [2]

It is not clear how to compare to [2] fairly since it does not provide methods for learning hyperparameters and their framework does not support such an extension. Accurate hyperparameter learning is required for real datasets like those in the paper. So [2] performs extremely poorly unless suitable settings for the hyperparameters can be guessed from the first batch of data. Furthermore, our paper goes beyond [2] by providing a method for optimising pseudo-inputs which has been shown to substantially improve upon the heuristics used in [2] in the batch setting [3].

### 1.3 Are SVI or the stream-based method performing differently due to different approximations?

No. Conventional SVI is fundamentally unsuited to the streaming setting and it performs very poorly practically compared to both the collapsed and uncollapsed versions of our method. The SVI learning rates require a lot of dataset and iteration specific tuning so the new data can be revisited multiple times without forgetting old data. The uncollapsed versions of our method do not require tuning of this sort and perform just as well as the collapsed version given sufficient updates.

### 1.4 Are pseudo-points appropriate for streaming settings?

In any setting (batch/streaming), pseudo-point approximations require the pseudo-points to cover the input space occupied by the data. This means they can be inappropriate for very long time-series or very high-dimensional inputs. This is a general issue with the approximation class. The development of new pseudo-point approximations to handle very large numbers of pseudo-points is a key and active research area [4], but orthogonal to our focus in this paper. A moving window could be introduced so just recent data are modelled (as we use for SGP/GP) but the utility of this depends on the task. Here we assume all input regions must be modelled which is problematic for windowing.

---

[*]These authors contributed equally to this work.

## 1.5 A possible explanation on why all models including full GP regression tend to learn bigger noise variances

This is a bias that arises because the learned functions are more discrepant from the training data than the true function and so the learned observation noise inflates to accommodate the mismatch.

## 1.6 Are the hyperparameters learned in the time-series and spatial data experiments?

Yes, hyperparameters and pseudo-inputs are optimised using the online variational free energy. This is absolutely central to our approach and the key difference to [2, 5].

## 1.7 Why is there a non-monotonic behaviour in fig. 4 in the main text?

This occurs because at some point the GP/SGP memory window cannot cover all observed data. Some parts of the input space are then missed, leading to decreasing performance.

# 2 Variational free energy approach for streaming sparse GP regression

## 2.1 The variational lower bound

Let $\mathbf{a} = f(\mathbf{z}_{\mathrm{old}})$ and $\mathbf{b} = f(\mathbf{z}_{\mathrm{new}})$ be the function values at the pseudo-inputs before and after seeing new data. The previous posterior, $q_{\mathrm{old}}(f) = p(f_{\neq \mathbf{a}}|\mathbf{a}, \theta_{\mathrm{old}})q(\mathbf{a})$, can be used to find the approximate likelihood given by old observations as follows,

$$p(\mathbf{y}_{\mathrm{old}}|f) \approx \frac{q_{\mathrm{old}}(f)p(\mathbf{y}_{\mathrm{old}}|\theta_{\mathrm{old}})}{p(f|\theta_{\mathrm{old}})} \quad \text{as} \quad q_{\mathrm{old}}(f) \approx \frac{p(f|\theta_{\mathrm{old}})p(\mathbf{y}_{\mathrm{old}}|f)}{p(\mathbf{y}_{\mathrm{old}}|\theta_{\mathrm{old}})}. \tag{1}$$

Substituting this into the posterior that we want to target gives us:

$$p(f|\mathbf{y}_{\mathrm{old}}, \mathbf{y}_{\mathrm{new}}) = \frac{p(f|\theta_{\mathrm{new}})p(\mathbf{y}_{\mathrm{old}}|f)p(\mathbf{y}_{\mathrm{new}}|f)}{p(\mathbf{y}_{\mathrm{new}}, \mathbf{y}_{\mathrm{old}}|\theta_{\mathrm{new}})} \approx \frac{p(f|\theta_{\mathrm{new}})q_{\mathrm{old}}(f)p(\mathbf{y}_{\mathrm{old}}|\theta_{\mathrm{old}})p(\mathbf{y}_{\mathrm{new}}|f)}{p(f|\theta_{\mathrm{old}})p(\mathbf{y}_{\mathrm{new}}, \mathbf{y}_{\mathrm{old}}|\theta_{\mathrm{new}})}.$$

The new posterior approximation takes the same form, but with the new pseudo-points and new hyperparameters: $q_{\mathrm{new}}(f) = p(f_{\neq \mathbf{b}}|\mathbf{b}, \theta_{\mathrm{new}})q(\mathbf{b})$. This approximate posterior can be obtained by minimising the KL divergence,

$$\mathrm{KL}[q_{\mathrm{new}}(f)||\hat{p}(f|\mathbf{y}_{\mathrm{old}}, \mathbf{y}_{\mathrm{new}})] = \int \mathrm{d}f q_{\mathrm{new}}(f) \log \frac{p(f_{\neq \mathbf{b}}|\mathbf{b}, \theta_{\mathrm{new}})q_{\mathrm{new}}(\mathbf{b})}{\frac{\mathcal{Z}_1(\theta_{\mathrm{old}})}{\mathcal{Z}_2(\theta_{\mathrm{new}})}p(f|\theta_{\mathrm{new}})p(\mathbf{y}_{\mathrm{new}}|f)\frac{q_{\mathrm{old}}(f)}{p(f|\theta_{\mathrm{old}})}} \tag{2}$$

$$= \log \frac{\mathcal{Z}_2(\theta_{\mathrm{new}})}{\mathcal{Z}_1(\theta_{\mathrm{old}})} + \int \mathrm{d}f q_{\mathrm{new}}(f) \left[ \log \frac{p(\mathbf{a}|\theta_{\mathrm{old}})q_{\mathrm{new}}(\mathbf{b})}{p(\mathbf{b}|\theta_{\mathrm{new}})q_{\mathrm{old}}(\mathbf{a})p(\mathbf{y}_{\mathrm{new}}|f)} \right]. \tag{3}$$

The last equation above is obtained by noting that $p(f|\theta_{\mathrm{new}})/p(f_{\neq \mathbf{b}}|\mathbf{b}, \theta_{\mathrm{new}}) = p(\mathbf{b}|\theta_{\mathrm{new}})$ and

$$\frac{q_{\mathrm{old}}(f)}{p(f|\theta_{\mathrm{old}})} = \frac{\overline{p(f_{\neq \mathbf{a}}|\mathbf{a}, \theta_{\mathrm{old}})}q_{\mathrm{old}}(\mathbf{a})}{\overline{p(f_{\neq \mathbf{a}}|\mathbf{a}, \theta_{\mathrm{old}})}p(\mathbf{a}|\theta_{\mathrm{old}})} = \frac{q_{\mathrm{old}}(\mathbf{a})}{p(\mathbf{a}|\theta_{\mathrm{old}})}.$$

Since the KL divergence is non-negative, the second term in (3) is the negative lower bound of the approximate online log marginal likelihood, or the variational free energy, $\mathcal{F}(q_{\mathrm{new}}(f))$. We can decompose the bound as follows,

$$\mathcal{F}(q_{\mathrm{new}}(f)) = \int \mathrm{d}f q_{\mathrm{new}}(f) \left[ \log \frac{p(\mathbf{a}|\theta_{\mathrm{old}})q_{\mathrm{new}}(\mathbf{b})}{p(\mathbf{b}|\theta_{\mathrm{new}})q_{\mathrm{old}}(\mathbf{a})p(\mathbf{y}_{\mathrm{new}}|f)} \right] \tag{4}$$

$$= \mathrm{KL}(q(\mathbf{b})||p(\mathbf{b}|\theta_{\mathrm{new}})) - \int q_{\mathrm{new}}(f) \log p(\mathbf{y}_{\mathrm{new}}|f)$$

$$+ \mathrm{KL}(q_{\mathrm{new}}(\mathbf{a})||q_{\mathrm{old}}(a)) - \mathrm{KL}(q_{\mathrm{new}}(\mathbf{a})||p(\mathbf{a}|\theta_{\mathrm{old}})). \tag{5}$$

The first two terms form the batch variational bound if the current batch is the whole training data, and the last two terms constrain the posterior to take into account the old likelihood (through the approximate posterior and the prior).

## 2.2 Derivation of the optimal posterior update and the collapsed bound

The aim is to find the new approximate posterior $q_{\text{new}}(f)$ such that the free energy is minimised. This is achieved by setting the derivative of $\mathcal{F}$ and a Lagrange term [2] w.r.t. $q(\mathbf{b})$ equal 0,

$$\frac{d\mathcal{F}}{dq(\mathbf{b})} + \lambda = \int df_{\neq \mathbf{b}} p(f_{\neq \mathbf{b}}|\mathbf{b}) \left[ \log \frac{p(\mathbf{a}|\theta_{\text{old}})q(\mathbf{b})}{p(\mathbf{b}|\theta_{\text{new}})q(\mathbf{a})} - \log p(\mathbf{y}|\mathbf{f}) \right] + 1 + \lambda = 0, \quad (6)$$

resulting in,

$$q_{\text{opt}}(\mathbf{b}) = \frac{1}{\mathcal{C}} p(\mathbf{b}) \exp \left( \int d\mathbf{a} p(\mathbf{a}|\mathbf{b}) \log \frac{q(\mathbf{a})}{p(\mathbf{a}|\theta_{\text{old}})} + \int d\mathbf{f} p(\mathbf{f}|\mathbf{b}) \log p(\mathbf{y}|\mathbf{f}) \right). \quad (7)$$

Note that we have dropped $\theta_{\text{new}}$ from $p(\mathbf{b}|\theta_{\text{new}})$, $p(\mathbf{a}|\mathbf{b}, \theta_{\text{new}})$ and $p(\mathbf{f}|\mathbf{b}, \theta_{\text{new}})$ to lighten the notation. Substituting the above result into the variational free energy leads to $\mathcal{F}(q_{\text{opt}}(f)) = -\log \mathcal{C}$. We now consider the exponents in the optimal $q_{\text{opt}}(\mathbf{b})$, noting that $q(\mathbf{a}) = \mathcal{N}(\mathbf{a}; \mathbf{m_a}, \mathbf{S_a})$ and $p(\mathbf{a}|\theta_{\text{old}}) = \mathcal{N}(\mathbf{a}; \mathbf{0}, \mathbf{K'_{aa}})$, and denoting $\mathbf{D_a} = (\mathbf{S_a}^{-1} - \mathbf{K'}_{\mathbf{aa}}^{-1})^{-1}$, $\mathbf{Q_f} = \mathbf{K_{ff}} - \mathbf{K_{fb}}\mathbf{K}_{\mathbf{bb}}^{-1}\mathbf{K_{bf}}$, and $\mathbf{Q_a} = \mathbf{K_{aa}} - \mathbf{K_{ab}}\mathbf{K}_{\mathbf{bb}}^{-1}\mathbf{K_{ba}}$,

$$E_1 = \int d\mathbf{a} p(\mathbf{a}|\mathbf{b}) \log \frac{q(\mathbf{a})}{p(\mathbf{a}|\theta_{\text{old}})} \quad (8)$$

$$= \frac{1}{2} \int d\mathbf{a} \mathcal{N}(\mathbf{a}; \mathbf{K_{ab}}\mathbf{K}_{\mathbf{bb}}^{-1}\mathbf{b}, \mathbf{Q_a}) \left( -\log \frac{|\mathbf{S_a}|}{|\mathbf{K'_{aa}}|} - (\mathbf{a} - \mathbf{m_a})^\mathsf{T}\mathbf{S_a}^{-1}(\mathbf{a} - \mathbf{m_a}) + \mathbf{a}^\mathsf{T}\mathbf{K'}_{\mathbf{aa}}^{-1}\mathbf{a} \right)$$

$$= \log \mathcal{N}(\mathbf{D_a}\mathbf{S_a}^{-1}\mathbf{m_a}; \mathbf{K_{ab}}\mathbf{K}_{\mathbf{bb}}^{-1}\mathbf{b}, \mathbf{D_a}) + \Delta_1, \quad (9)$$

$$E_2 = \int d\mathbf{f} p(\mathbf{f}|\mathbf{b}) \log p(\mathbf{y}|\mathbf{f}) \quad (10)$$

$$= \int d\mathbf{f} \mathcal{N}(\mathbf{f}; \mathbf{K_{fb}}\mathbf{K}_{\mathbf{bb}}^{-1}\mathbf{b}, \mathbf{Q_f}) \log \mathcal{N}(\mathbf{y}; \mathbf{f}, \sigma^2\mathbf{I}) \quad (11)$$

$$= \log \mathcal{N}(\mathbf{y}; \mathbf{K_{fb}}\mathbf{K}_{\mathbf{bb}}^{-1}\mathbf{b}, \sigma^2\mathbf{I}) + \Delta_2, \quad (12)$$

$$2\Delta_1 = -\log \frac{|\mathbf{S_a}|}{|\mathbf{K'_{aa}}||\mathbf{D_a}|} + \mathbf{m_a}^\mathsf{T}\mathbf{S_a}^{-1}\mathbf{D_a}\mathbf{S_a}^{-1}\mathbf{m_a} - \text{tr}[\mathbf{D_a}^{-1}\mathbf{Q_a}] - \mathbf{m_a}^\mathsf{T}\mathbf{S_a}^{-1}\mathbf{m_a} + M_a \log(2\pi), \quad (13)$$

$$\Delta_2 = -\frac{1}{2\sigma^2}\text{tr}(\mathbf{Q_f}). \quad (14)$$

Putting these results back into the optimal $q(\mathbf{b})$, we obtain:

$$q_{\text{opt}}(\mathbf{b}) \propto p(\mathbf{b})\mathcal{N}(\hat{\mathbf{y}}, \mathbf{K_{\hat{f}b}}\mathbf{K}_{\mathbf{bb}}^{-1}\mathbf{b}, \Sigma_{\hat{\mathbf{y}}}) \quad (15)$$

$$= \mathcal{N}(\mathbf{b}; \mathbf{K_{b\hat{f}}}(\mathbf{K_{\hat{f}b}}\mathbf{K}_{\mathbf{bb}}^{-1}\mathbf{K_{b\hat{f}}} + \Sigma_{\hat{\mathbf{y}}})^{-1}\hat{\mathbf{y}}, \mathbf{K_{bb}} - \mathbf{K_{b\hat{f}}}(\mathbf{K_{\hat{f}b}}\mathbf{K}_{\mathbf{bb}}^{-1}\mathbf{K_{b\hat{f}}} + \Sigma_{\hat{\mathbf{y}}})^{-1}\mathbf{K_{\hat{f}b}}) \quad (16)$$

where

$$\hat{\mathbf{y}} = \begin{bmatrix} \mathbf{y} \\ \mathbf{D_a}\mathbf{S_a}^{-1}\mathbf{m_a} \end{bmatrix}, \quad \mathbf{K_{\hat{f}b}} = \begin{bmatrix} \mathbf{K_{fb}} \\ \mathbf{K_{ab}} \end{bmatrix}, \quad \Sigma_{\hat{\mathbf{y}}} = \begin{bmatrix} \sigma_y^2\mathbf{I} & \mathbf{0} \\ \mathbf{0} & \mathbf{D_a} \end{bmatrix}. \quad (17)$$

The negative variational free energy, which is the lower bound of the log marginal likelihood, can also be derived,

$$\mathcal{F} = \log \mathcal{C} = \log \mathcal{N}(\hat{\mathbf{y}}; \mathbf{0}, \mathbf{K_{\hat{f}b}}\mathbf{K}_{\mathbf{bb}}^{-1}\mathbf{K_{b\hat{f}}} + \Sigma_{\hat{\mathbf{y}}}) + \Delta_1 + \Delta_2. \quad (18)$$

## 2.3 Implementation

In this section, we provide efficient and numerical stable forms for a practical implementation of the above results.

### 2.3.1 The variational free energy

The first term in eq. (18) can be written as follows,

$$\mathcal{F}_1 = \log \mathcal{N}(\hat{\mathbf{y}}; \mathbf{0}, \mathbf{K_{\hat{f}b}}\mathbf{K}_{\mathbf{bb}}^{-1}\mathbf{K_{b\hat{f}}} + \Sigma_{\hat{\mathbf{y}}}) \quad (19)$$

$$= -\frac{N + M_a}{2} \log(2\pi) - \frac{1}{2}\log|\mathbf{K_{\hat{f}b}K_{bb}^{-1}K_{b\hat{f}}} + \Sigma_{\hat{y}}| - \frac{1}{2}\hat{\mathbf{y}}^\intercal(\mathbf{K_{\hat{f}b}K_{bb}^{-1}K_{b\hat{f}}} + \Sigma_{\hat{y}})^{-1}\hat{\mathbf{y}}. \quad (20)$$

Let $\mathbf{S}_y = \mathbf{K_{\hat{f}b}K_{bb}^{-1}K_{b\hat{f}}} + \Sigma_{\hat{y}}$ and $\mathbf{K_{bb}} = \mathbf{L_b L_b^\intercal}$, using the matrix determinant lemma, we obtain,

$$\log|\mathbf{S_y}| = \log|\mathbf{K_{\hat{f}b}K_{bb}^{-1}K_{b\hat{f}}} + \Sigma_{\hat{y}}| \quad (21)$$

$$= \log|\Sigma_{\hat{y}}| + \log|\mathbf{I} + \mathbf{L_b^{-1}K_{b\hat{f}}}\Sigma_{\hat{y}}^{-1}\mathbf{K_{\hat{f}b}L_b^{-\intercal}}| \quad (22)$$

$$= N\log\sigma_y^2 + \log|\mathbf{D_a}| + \log|\mathbf{I} + \mathbf{L_b^{-1}K_{b\hat{f}}}\Sigma_{\hat{y}}^{-1}\mathbf{K_{\hat{f}b}L_b^{-\intercal}}|. \quad (23)$$

Let $\mathbf{D} = \mathbf{I} + \mathbf{L_b^{-1}K_{b\hat{f}}}\Sigma_{\hat{y}}^{-1}\mathbf{K_{\hat{f}b}L_b^{-\intercal}}$. Note that,

$$\mathbf{K_{b\hat{f}}}\Sigma_{\hat{y}}^{-1}\mathbf{K_{\hat{f}b}} = \frac{1}{\sigma_y^2}\mathbf{K_{bf}K_{fb}} + \mathbf{K_{ba}S_a^{-1}K_{ab}} - \mathbf{K_{ba}K_{aa}'^{-1}K_{ab}}. \quad (24)$$

Using the matrix inversion lemma gives us,

$$\mathbf{S_y^{-1}} = (\mathbf{K_{\hat{f}b}K_{bb}^{-1}K_{b\hat{f}}} + \Sigma_{\hat{y}})^{-1} \quad (25)$$

$$= \Sigma_{\hat{y}}^{-1} - \Sigma_{\hat{y}}^{-1}\mathbf{K_{\hat{f}b}L_b^{-\intercal}D^{-1}L_b^{-1}K_{b\hat{f}}}\Sigma_{\hat{y}}^{-1}, \quad (26)$$

leading to,

$$\hat{\mathbf{y}}^\intercal\mathbf{S_y^{-1}}\hat{\mathbf{y}} = \hat{\mathbf{y}}^\intercal\Sigma_{\hat{y}}^{-1}\hat{\mathbf{y}} - \hat{\mathbf{y}}^\intercal\Sigma_{\hat{y}}^{-1}\mathbf{K_{\hat{f}b}L_b^{-\intercal}D^{-1}L_b^{-1}K_{b\hat{f}}}\Sigma_{\hat{y}}^{-1}\hat{\mathbf{y}}. \quad (27)$$

Note that,

$$\hat{\mathbf{y}}^\intercal\Sigma_{\hat{y}}^{-1}\hat{\mathbf{y}} = \frac{1}{\sigma_y^2}\mathbf{y}^\intercal\mathbf{y} + \mathbf{m_a^\intercal S_a^{-1}D_a S_a^{-1}m_a}, \quad (28)$$

$$\text{and} \quad \mathbf{c} = \mathbf{K_{b\hat{f}}}\Sigma_{\hat{y}}^{-1}\hat{\mathbf{y}} = \frac{1}{\sigma^2}\mathbf{K_{bf}y} + \mathbf{K_{ba}S_a^{-1}m_a}. \quad (29)$$

Substituting these results back into equation eq. (18),

$$\mathcal{F} = -\frac{N}{2}\log(2\pi\sigma^2) - \frac{1}{2}\log|\mathbf{D}| - \frac{1}{2\sigma^2}\mathbf{y}^\intercal\mathbf{y} + \frac{1}{2}\mathbf{c}^\intercal\mathbf{L_b^{-\intercal}D^{-1}L_b^{-1}c}$$
$$- \frac{1}{2}\log|\mathbf{S_a}| + \frac{1}{2}\log|\mathbf{K_{aa}'}| - \frac{1}{2}\text{tr}[\mathbf{D_a^{-1}Q_a}] - \frac{1}{2}\mathbf{m_a^\intercal S_a^{-1}m_a} - \frac{1}{2\sigma^2}\text{tr}(\mathbf{Q_f}). \quad (30)$$

### 2.3.2 Prediction

We revisit and rewrite the optimal variational distribution, $q_{\text{opt}}(\mathbf{b})$, using its natural parameters:

$$q_{\text{opt}}(\mathbf{b}) \propto p(\mathbf{b})\mathcal{N}(\hat{\mathbf{y}}, \mathbf{K_{\hat{f}b}K_{bb}^{-1}b}, \Sigma_{\hat{y}}) \quad (31)$$

$$= \mathcal{N}^{-1}(\mathbf{b}; \mathbf{K_{bb}^{-1}K_{b\hat{f}}}\Sigma_{\hat{y}}^{-1}\hat{\mathbf{y}}, \mathbf{K_{bb}^{-1}} + \mathbf{K_{bb}^{-1}K_{b\hat{f}}}\Sigma_{\hat{y}}^{-1}\mathbf{K_{\hat{f}b}K_{bb}^{-1}}). \quad (32)$$

The predictive covariance at some test points $\mathbf{s}$ is:

$$\mathbf{V_{ss}} = \mathbf{K_{ss}} - \mathbf{K_{sb}K_{bb}^{-1}K_{bs}} + \mathbf{K_{sb}K_{bb}^{-1}}(\mathbf{K_{bb}^{-1}} + \mathbf{K_{bb}^{-1}K_{b\hat{f}}}\Sigma_{\hat{y}}^{-1}\mathbf{K_{\hat{f}b}K_{bb}^{-1}})^{-1}\mathbf{K_{bb}^{-1}K_{bs}} \quad (33)$$

$$= \mathbf{K_{ss}} - \mathbf{K_{sb}K_{bb}^{-1}K_{bs}} + \mathbf{K_{sb}L_b^{-\intercal}}(\mathbf{I} + \mathbf{L_b^{-1}K_{b\hat{f}}}\Sigma_{\hat{y}}^{-1}\mathbf{K_{\hat{f}b}L_b^{-\intercal}})^{-1}\mathbf{L_b^{-\intercal}K_{bs}} \quad (34)$$

$$= \mathbf{K_{ss}} - \mathbf{K_{sb}K_{bb}^{-1}K_{bs}} + \mathbf{K_{sb}L_b^{-\intercal}D^{-1}L_b^{-\intercal}K_{bs}}. \quad (35)$$

And the predictive mean is:

$$\mathbf{m_s} = \mathbf{K_{sb}K_{bb}^{-1}}(\mathbf{K_{bb}^{-1}} + \mathbf{K_{bb}^{-1}K_{b\hat{f}}}\Sigma_{\hat{y}}^{-1}\mathbf{K_{\hat{f}b}K_{bb}^{-1}})^{-1}\mathbf{K_{bb}^{-1}K_{b\hat{f}}}\Sigma_{\hat{y}}^{-1}\hat{\mathbf{y}} \quad (36)$$

$$= \mathbf{K_{sb}L_b^{-\intercal}}(\mathbf{I} + \mathbf{L_b^{-1}K_{b\hat{f}}}\Sigma_{\hat{y}}^{-1}\mathbf{K_{\hat{f}b}L_b^{-\intercal}})^{-1}\mathbf{L_b^{-1}K_{b\hat{f}}}\Sigma_{\hat{y}}^{-1}\hat{\mathbf{y}} \quad (37)$$

$$= \mathbf{K_{sb}L_b^{-\intercal}D^{-1}L_b^{-1}K_{b\hat{f}}}\Sigma_{\hat{y}}^{-1}\hat{\mathbf{y}}. \quad (38)$$

## 3 Power-EP for streaming sparse Gaussian process regression

Similar to the variational approach above, we also use $\mathbf{a} = f(\mathbf{z}_{\text{old}})$ and $\mathbf{b} = f(\mathbf{z}_{\text{new}})$ as pseudo-outputs before and after seeing new data. The exact posterior upon observing new data is

$$p(f|\mathbf{y}, \mathbf{y}_{\text{old}}) = \frac{1}{\mathcal{Z}}p(f_{\neq \mathbf{a}}|\mathbf{a}, \theta_{\text{old}})q(\mathbf{a})p(\mathbf{y}|f) \tag{39}$$

$$= \frac{1}{\mathcal{Z}}p(f|\theta_{\text{old}})\frac{q(\mathbf{a})}{p(\mathbf{a}|\theta_{\text{old}})}p(\mathbf{y}|f). \tag{40}$$

In addition, we assume that the hyperparameters do not change significantly after each online update and as a result, the exact posterior can be approximated by:

$$p(f|\mathbf{y}, \mathbf{y}_{\text{old}}) \approx \frac{1}{\mathcal{Z}}p(f|\theta_{\text{new}})\frac{q(\mathbf{a})}{p(\mathbf{a}|\theta_{\text{old}})}p(\mathbf{y}|f). \tag{41}$$

We posit the following approximate posterior, which mirrors the form of the exact posterior,

$$q(f) \propto p(f|\theta_{\text{new}})q_1(\mathbf{b})q_2(\mathbf{b}), \tag{42}$$

where $q_1(\mathbf{b})$ and $q_2(\mathbf{b})$ are the approximate effect that $\frac{q(\mathbf{a})}{p(\mathbf{a}|\theta_{\text{old}})}$ and $p(\mathbf{y}|f)$ have on the posterior, respectively. Next we describe steps to obtain the closed-form expressions for the approximate factors and the approximate marginal likelihood.

### 3.1 $q_1(\mathbf{b})$

The cavity and tilted distributions are:

$$q_{\text{cav},1}(f) = p(f)q_1^{1-\alpha}(\mathbf{b})q_2(\mathbf{b}) \tag{43}$$

$$= p(f_{\neq \mathbf{a},\mathbf{b}}|\mathbf{b})p(\mathbf{b})q_2(\mathbf{b})p(\mathbf{a}|\mathbf{b})q_1^{1-\alpha}(\mathbf{b}) \tag{44}$$

$$\text{and } \tilde{q}_1(f) = p(f_{\neq \mathbf{a},\mathbf{b}}|\mathbf{b})p(\mathbf{b})q_2(\mathbf{b})p(\mathbf{a}|\mathbf{b})q_1^{1-\alpha}(\mathbf{b})\left(\frac{q(\mathbf{a})}{p(\mathbf{a}|\theta_{\text{old}})}\right)^\alpha. \tag{45}$$

We note that, $q(\mathbf{a}) = \mathcal{N}(\mathbf{a}; \mathbf{m_a}, \mathbf{S_a})$ and $p(\mathbf{a}|\theta_{\text{old}}) = \mathcal{N}(a; 0, \mathbf{K'_{aa}})$, leading to:

$$\left(\frac{q(\mathbf{a})}{p(\mathbf{a}|\theta_{\text{old}})}\right)^\alpha = C_1 \mathcal{N}(\mathbf{a}; \hat{\mathbf{m}}_\mathbf{a}, \hat{\mathbf{S}}_\mathbf{a}) \tag{46}$$

$$\text{where } \hat{\mathbf{m}}_\mathbf{a} = \mathbf{D_a}\mathbf{S_a}^{-1}\mathbf{m_a}, \tag{47}$$

$$\hat{\mathbf{S}}_\mathbf{a} = \frac{1}{\alpha}\mathbf{D_a}, \tag{48}$$

$$\mathbf{D_a} = (\mathbf{S_a}^{-1} - \mathbf{K'_{aa}}^{-1})^{-1}, \tag{49}$$

$$C_1 = (2\pi)^{M/2}|\mathbf{K'_{aa}}|^{\alpha/2}|\mathbf{S_a}|^{-\alpha/2}|\hat{\mathbf{S}}_\mathbf{a}|^{1/2}\exp(\frac{\alpha}{2}\mathbf{m_a^\mathsf{T}}[\mathbf{S_a}^{-1}\mathbf{D_a}\mathbf{S_a}^{-1} - \mathbf{S_a}^{-1}]\mathbf{m_a}). \tag{50}$$

Let $\Sigma_\mathbf{a} = \mathbf{D_a} + \alpha\mathbf{Q_a}$. Note that:

$$p(\mathbf{a}|\mathbf{b}) = \mathcal{N}(\mathbf{a}; \mathbf{K_{ab}}\mathbf{K_{bb}}^{-1}\mathbf{b}; \mathbf{K_{aa}} - \mathbf{K_{ab}}\mathbf{K_{bb}}^{-1}\mathbf{K_{ba}}) = \mathcal{N}(\mathbf{a}; \mathbf{W_a}\mathbf{b}, \mathbf{Q_a}). \tag{51}$$

As a result,

$$\int \mathrm{d}\mathbf{a}\,p(\mathbf{a}|\mathbf{b})\left(\frac{q(\mathbf{a})}{p(\mathbf{a}|\theta_{\text{old}})}\right)^\alpha = \int \mathrm{d}\mathbf{a}\,C_1 \mathcal{N}(\mathbf{a}; \hat{\mathbf{m}}_\mathbf{a}, \hat{\mathbf{S}}_\mathbf{a})\mathcal{N}(\mathbf{a}; \mathbf{W_a}\mathbf{b}, \mathbf{Q_a}) \tag{52}$$

$$= C_1 \mathcal{N}(\hat{\mathbf{m}}_\mathbf{a}; \mathbf{W_a}\mathbf{b}, \Sigma_\mathbf{a}/\alpha). \tag{53}$$

Since this is the contribution towards the posterior from $\mathbf{a}$, it needs to match $q_1^\alpha(\mathbf{b})$ at convergence, that is,

$$q_1(\mathbf{b}) \propto [C_1 \mathcal{N}(\hat{\mathbf{m}}_\mathbf{a}; \mathbf{W_a}\mathbf{b}, \Sigma_\mathbf{a}/\alpha)]^{1/\alpha} \tag{54}$$

$$= \mathcal{N}(\hat{\mathbf{m}}_\mathbf{a}; \mathbf{W_a}\mathbf{b}, \alpha(\Sigma_\mathbf{a}/\alpha)) \tag{55}$$

$$= \mathcal{N}(\hat{\mathbf{m}}_\mathbf{a}; \mathbf{W_a}\mathbf{b}, \Sigma_\mathbf{a}). \tag{56}$$

In addition, we can compute:

$$\log \tilde{Z}_1 = \log \int \mathrm{d}f \tilde{q}_1(f) \tag{57}$$

$$= \log C_1 \mathcal{N}(\hat{\mathbf{m}}_{\mathbf{a}}; \mathbf{W}_{\mathbf{a}}\mathbf{m}_{\mathrm{cav}}, \Sigma_{\mathbf{a}}/\alpha + \mathbf{W}_{\mathbf{a}}\mathbf{V}_{\mathrm{cav}}\mathbf{W}_{\mathbf{a}}^{\intercal}) \tag{58}$$

$$= \log C_1 - \frac{M}{2}\log(2\pi) - \frac{1}{2}\log|\Sigma_{\mathbf{a}}/\alpha + \mathbf{W}_{\mathbf{a}}\mathbf{V}_{\mathrm{cav}}\mathbf{W}_{\mathbf{a}}^{\intercal}| - \frac{1}{2}\hat{\mathbf{m}}_{\mathbf{a}}^{\intercal}(\Sigma_{\mathbf{a}}/\alpha + \mathbf{W}_{\mathbf{a}}\mathbf{V}_{\mathrm{cav}}\mathbf{W}_{\mathbf{a}}^{\intercal})^{-1}\hat{\mathbf{m}}_{\mathbf{a}}$$
$$+ \mathbf{m}_{\mathrm{cav}}^{\intercal}\mathbf{W}_{\mathbf{a}}^{\intercal}(\Sigma_{\mathbf{a}}/\alpha + \mathbf{W}_{\mathbf{a}}\mathbf{V}_{\mathrm{cav}}\mathbf{W}_{\mathbf{a}}^{\intercal})^{-1}\hat{\mathbf{m}}_{\mathbf{a}} - \frac{1}{2}\mathbf{m}_{\mathrm{cav}}^{\intercal}\mathbf{W}_{\mathbf{a}}^{\intercal}(\Sigma_{\mathbf{a}}/\alpha + \mathbf{W}_{\mathbf{a}}\mathbf{V}_{\mathrm{cav}}\mathbf{W}_{\mathbf{a}}^{\intercal})^{-1}\mathbf{W}_{\mathbf{a}}\mathbf{m}_{\mathrm{cav}}. \tag{59}$$

Note that:

$$\mathbf{V}^{-1} = \mathbf{V}_{\mathrm{cav}}^{-1} + \mathbf{W}_{\mathbf{a}}^{\intercal}(\Sigma_{\mathbf{a}}/\alpha)^{-1}\mathbf{W}_{\mathbf{a}}, \tag{60}$$

$$\mathbf{V}^{-1}\mathbf{m} = \mathbf{V}_{\mathrm{cav}}^{-1}\mathbf{m}_{\mathrm{cav}} + \mathbf{W}_{\mathbf{a}}^{\intercal}(\Sigma_{\mathbf{a}}/\alpha)^{-1}\hat{\mathbf{m}}_{\mathbf{a}}. \tag{61}$$

Using matrix inversion lemma gives

$$\mathbf{V} = \mathbf{V}_{\mathrm{cav}} - \mathbf{V}_{\mathrm{cav}}\mathbf{W}_{\mathbf{a}}^{\intercal}(\Sigma_{\mathbf{a}}/\alpha + \mathbf{W}_{\mathbf{a}}\mathbf{V}_{\mathrm{cav}}\mathbf{W}_{\mathbf{a}}^{\intercal})^{-1}\mathbf{W}_{\mathbf{a}}\mathbf{V}_{\mathrm{cav}}. \tag{62}$$

Using matrix determinant lemma gives

$$|\mathbf{V}^{-1}| = |\mathbf{V}_{\mathrm{cav}}^{-1}||(\Sigma_{\mathbf{a}}/\alpha)^{-1}||\Sigma_{\mathbf{a}}/\alpha + \mathbf{W}_{\mathbf{a}}\mathbf{V}_{\mathrm{cav}}\mathbf{W}_{\mathbf{a}}^{\intercal}|. \tag{63}$$

We can expand terms in $\log \tilde{Z}_1$ above as follows:

$$\log \tilde{Z}_{1A} = -\frac{1}{2}\log|\Sigma_{\mathbf{a}}/\alpha + \mathbf{W}_{\mathbf{a}}\mathbf{V}_{\mathrm{cav}}\mathbf{W}_{\mathbf{a}}^{\intercal}| \tag{64}$$

$$= -\frac{1}{2}(\log|\mathbf{V}^{-1}| - \log|\mathbf{V}_{\mathrm{cav}}^{-1}| - \log|(\Sigma_{\mathbf{a}}/\alpha)^{-1}|) \tag{65}$$

$$= \frac{1}{2}\log|\mathbf{V}| - \frac{1}{2}\log|\mathbf{V}_{\mathrm{cav}}| - \frac{1}{2}\log|(\Sigma_{\mathbf{a}}/\alpha)|. \tag{66}$$

$$\log \tilde{Z}_{1B} = -\frac{1}{2}\hat{\mathbf{m}}_{\mathbf{a}}^{\intercal}(\Sigma_{\mathbf{a}}/\alpha + \mathbf{W}_{\mathbf{a}}\mathbf{V}_{\mathrm{cav}}\mathbf{W}_{\mathbf{a}}^{\intercal})^{-1}\hat{\mathbf{m}}_{\mathbf{a}} \tag{67}$$

$$= -\frac{1}{2}\hat{\mathbf{m}}_{\mathbf{a}}^{\intercal}(\Sigma_{\mathbf{a}}/\alpha)^{-1}\hat{\mathbf{m}}_{\mathbf{a}} + \frac{1}{2}\hat{\mathbf{m}}_{\mathbf{a}}^{\intercal}(\Sigma_{\mathbf{a}}/\alpha)^{-1}\mathbf{W}_{\mathbf{a}}\mathbf{V}\mathbf{W}_{\mathbf{a}}^{\intercal}(\Sigma_{\mathbf{a}}/\alpha)^{-1}\hat{\mathbf{m}}_{\mathbf{a}}. \tag{68}$$

$$\log \tilde{Z}_{1C} = \mathbf{m}_{\mathrm{cav}}^{\intercal}\mathbf{W}_{\mathbf{a}}^{\intercal}(\Sigma_{\mathbf{a}}/\alpha + \mathbf{W}_{\mathbf{a}}\mathbf{V}_{\mathrm{cav}}\mathbf{W}_{\mathbf{a}}^{\intercal})^{-1}\hat{\mathbf{m}}_{\mathbf{a}} \tag{69}$$

$$= \mathbf{m}_{\mathrm{cav}}^{\intercal}\mathbf{W}_{\mathbf{a}}^{\intercal}(\Sigma_{\mathbf{a}}/\alpha)^{-1}\hat{\mathbf{m}}_{\mathbf{a}} - \mathbf{m}_{\mathrm{cav}}^{\intercal}\mathbf{W}_{\mathbf{a}}^{\intercal}(\Sigma_{\mathbf{a}}/\alpha)^{-1}\mathbf{W}_{\mathbf{a}}\mathbf{V}\mathbf{W}_{\mathbf{a}}^{\intercal}(\Sigma_{\mathbf{a}}/\alpha)^{-1}\hat{\mathbf{m}}_{\mathbf{a}}. \tag{70}$$

$$\log \tilde{Z}_{1D} = -\frac{1}{2}\mathbf{m}_{\mathrm{cav}}^{\intercal}\mathbf{W}_{\mathbf{a}}^{\intercal}(\Sigma_{\mathbf{a}}/\alpha + \mathbf{W}_{\mathbf{a}}\mathbf{V}_{\mathrm{cav}}\mathbf{W}_{\mathbf{a}}^{\intercal})^{-1}\mathbf{W}_{\mathbf{a}}\mathbf{m}_{\mathrm{cav}} \tag{71}$$

$$= -\frac{1}{2}\mathbf{m}_{\mathrm{cav}}^{\intercal}\mathbf{V}_{\mathrm{cav}}^{-1}\mathbf{m}_{\mathrm{cav}} + \frac{1}{2}\mathbf{m}_{\mathrm{cav}}^{\intercal}\mathbf{V}_{\mathrm{cav}}^{-1}\mathbf{V}\mathbf{V}_{\mathrm{cav}}^{-1}\mathbf{m}_{\mathrm{cav}} \tag{72}$$

$$= -\frac{1}{2}\mathbf{m}_{\mathrm{cav}}^{\intercal}\mathbf{V}_{\mathrm{cav}}^{-1}\mathbf{m}_{\mathrm{cav}} + \frac{1}{2}\mathbf{m}^{\intercal}\mathbf{V}^{-1}\mathbf{m}$$
$$+ \frac{1}{2}\hat{\mathbf{m}}_{\mathbf{a}}^{\intercal}(\Sigma_{\mathbf{a}}/\alpha)^{-1}\mathbf{W}_{\mathbf{a}}\mathbf{V}\mathbf{W}_{\mathbf{a}}^{\intercal}(\Sigma_{\mathbf{a}}/\alpha)^{-1}\hat{\mathbf{m}}_{\mathbf{a}} - \hat{\mathbf{m}}_{\mathbf{a}}(\Sigma_{\mathbf{a}}/\alpha)^{-1}\mathbf{W}_{\mathbf{a}}\mathbf{m}. \tag{73}$$

$$\log \tilde{Z}_{1DA} = -\hat{\mathbf{m}}_{\mathbf{a}}(\Sigma_{\mathbf{a}}/\alpha)^{-1}\mathbf{W}_{\mathbf{a}}\mathbf{m} \tag{74}$$

$$= -\hat{\mathbf{m}}_{\mathbf{a}}^{\intercal}(\Sigma_{\mathbf{a}}/\alpha)^{-1}\mathbf{W}_{\mathbf{a}}\mathbf{V}\mathbf{V}_{\mathrm{cav}}^{-1}\mathbf{m}_{\mathrm{cav}} - \hat{\mathbf{m}}_{\mathbf{a}}^{\intercal}(\Sigma_{\mathbf{a}}/\alpha)^{-1}\mathbf{W}_{\mathbf{a}}\mathbf{V}\mathbf{W}_{\mathbf{a}}^{\intercal}(\Sigma_{\mathbf{a}}/\alpha)^{-1}\hat{\mathbf{m}}_{\mathbf{a}}. \tag{75}$$

$$\log \tilde{Z}_{1DA1} = -\hat{\mathbf{m}}_{\mathbf{a}}^{\intercal}((\Sigma_{\mathbf{a}}/\alpha)^{-1})\mathbf{W}_{\mathbf{a}}\mathbf{V}\mathbf{V}_{\mathrm{cav}}^{-1}\mathbf{m}_{\mathrm{cav}} \tag{76}$$

$$= -\hat{\mathbf{m}}_{\mathbf{a}}^{\intercal}(\Sigma_{\mathbf{a}}/\alpha)^{-1}\mathbf{W}_{\mathbf{a}}(\mathbf{I} - \mathbf{V}\mathbf{W}_{\mathbf{a}}^{\intercal}\Sigma_{\mathbf{a}}/\alpha)^{-1}\mathbf{W}_{\mathbf{a}})\mathbf{m}_{\mathrm{cav}} \tag{77}$$

$$= -\mathbf{m}_{\mathrm{cav}}^{\intercal}\mathbf{W}_{\mathbf{a}}^{\intercal}(\Sigma_{\mathbf{a}}/\alpha)^{-1}\hat{\mathbf{m}}_{\mathbf{a}} + \mathbf{m}_{\mathrm{cav}}^{\intercal}\mathbf{W}_{\mathbf{a}}^{\intercal}(\Sigma_{\mathbf{a}}/\alpha)^{-1}\mathbf{W}_{\mathbf{a}}\mathbf{V}\mathbf{W}_{\mathbf{a}}^{\intercal}(\Sigma_{\mathbf{a}}/\alpha)^{-1}\hat{\mathbf{m}}_{\mathbf{a}}. \tag{78}$$

which results in:

$$\log \tilde{Z}_1 + \phi_{\mathrm{cav},1} - \phi_{\mathrm{post}} = \log C_1 - \frac{M}{2}\log(2\pi) - \frac{1}{2}\log|(\Sigma_{\mathbf{a}}/\alpha)| - \frac{1}{2}\hat{\mathbf{m}}_{\mathbf{a}}^{\intercal}(\Sigma_{\mathbf{a}}/\alpha)^{-1}\hat{\mathbf{m}}_{\mathbf{a}}. \tag{79}$$

## 3.2  $q_2(\mathbf{b})$

We repeat the above procedure to find $q_2(\mathbf{b})$. The cavity and tilted distributions are,

$$q_{\mathrm{cav},2}(f) = p(f)q_1(\mathbf{b})q_2^{1-\alpha}(\mathbf{b}) \tag{80}$$

$$= p(f_{\neq \mathbf{f},\mathbf{b}|\mathbf{b})p(\mathbf{b})q_1(\mathbf{b})p(\mathbf{f}|\mathbf{b})}q_2^{1-\alpha}(\mathbf{b}) \tag{81}$$

$$\text{and } \tilde{q}_2(f) = p(f_{\neq \mathbf{f},\mathbf{b}}|\mathbf{b})p(\mathbf{b})q_1(\mathbf{b})p(\mathbf{a}|\mathbf{b})q_2^{1-\alpha}(\mathbf{b})p^\alpha(\mathbf{y}|\mathbf{f}) \tag{82}$$

We note that, $p(\mathbf{y}|\mathbf{f}) = \mathcal{N}(\mathbf{y}; \mathbf{f}, \sigma_y^2 \mathbf{I})$ leading to,

$$p^\alpha(\mathbf{y}|\mathbf{f}) = C_2 \mathcal{N}(\mathbf{y}; \mathbf{f}, \hat{\mathbf{S}}_{\mathbf{y}}) \tag{83}$$

$$\text{where } \hat{\mathbf{S}}_{\mathbf{y}} = \frac{\sigma_y^2}{\alpha}\mathbf{I} \tag{84}$$

$$C_2 = (2\pi\sigma_y^2)^{N(1-\alpha)/2}\alpha^{-N/2} \tag{85}$$

Let $\Sigma_{\mathbf{y}} = \sigma_y^2 \mathbf{I} + \alpha \mathbf{Q_f}$. Note that,

$$p(\mathbf{f}|\mathbf{b}) = \mathcal{N}(\mathbf{f}; \mathbf{K_{fb}}\mathbf{K_{bb}^{-1}}\mathbf{b}; \mathbf{K_{ff}} - \mathbf{K_{fb}}\mathbf{K_{bb}^{-1}}\mathbf{K_{bf}}) = \mathcal{N}(\mathbf{a}; \mathbf{W_f}\mathbf{b}, \mathbf{Q_f}) \tag{86}$$

As a result,

$$\int \mathrm{d}\mathbf{a}\, p(\mathbf{f}|\mathbf{b})p^\alpha(\mathbf{y}|\mathbf{f}) = \int \mathrm{d}\mathbf{f}\, C_2 \mathcal{N}(\mathbf{y}; \mathbf{f}, \hat{\mathbf{S}}_{\mathbf{y}})\mathcal{N}(\mathbf{f}; \mathbf{W_f}\mathbf{b}, B) \tag{87}$$

$$= C_2 \mathcal{N}(\mathbf{y}; \mathbf{W_f}\mathbf{b}, \hat{\mathbf{S}}_{\mathbf{y}} + \mathbf{Q_f}) \tag{88}$$

Since this is the contribution towards the posterior from $\mathbf{y}$, it needs to match $q^\alpha(\mathbf{b})$ at convergence, that is,

$$q_2(\mathbf{b}) \propto \left[ C_2 \mathcal{N}(\mathbf{y}; \mathbf{W_f}\mathbf{b}, \hat{\mathbf{S}}_{\mathbf{y}} + \mathbf{Q_f}) \right]^{1/\alpha} \tag{89}$$

$$= \mathcal{N}(\mathbf{y}; \mathbf{W_f}\mathbf{b}, \alpha(\Sigma_{\mathbf{y}}/\alpha)) \tag{90}$$

$$= \mathcal{N}(\mathbf{y}; \mathbf{W_f}\mathbf{b}, \Sigma_{\mathbf{y}}) \tag{91}$$

In addition, we can compute,

$$\log \tilde{Z}_2 = \log \int \mathrm{d}f\, \tilde{q}_2(f) \tag{92}$$

$$= \log C_2 \mathcal{N}(\mathbf{y}; \mathbf{W_f}\mathbf{m}_{\mathrm{cav}}, \Sigma_{\mathbf{y}}/\alpha + \mathbf{W_f}\mathbf{V}_{\mathrm{cav}}\mathbf{W_f^\mathsf{T}}) \tag{93}$$

$$= \log C_2 - \frac{N}{2}\log(2\pi) - \frac{1}{2}\log|\Sigma_{\mathbf{y}}/\alpha + \mathbf{W_f}\mathbf{V}_{\mathrm{cav}}\mathbf{W_f^\mathsf{T}}| - \frac{1}{2}\mathbf{y}^\mathsf{T}(\Sigma_{\mathbf{y}}/\alpha + \mathbf{W_f}\mathbf{V}_{\mathrm{cav}}\mathbf{W_f^\mathsf{T}})^{-1}\mathbf{y}$$
$$+ \mathbf{m}_{\mathrm{cav}}^\mathsf{T}\mathbf{W_f^\mathsf{T}}(\Sigma_{\mathbf{y}}/\alpha + \mathbf{W_f}\mathbf{V}_{\mathrm{cav}}\mathbf{W_f^\mathsf{T}})^{-1}\mathbf{y} - \frac{1}{2}\mathbf{m}_{\mathrm{cav}}^\mathsf{T}\mathbf{W_f^\mathsf{T}}(\Sigma_{\mathbf{y}}/\alpha + \mathbf{W_f}\mathbf{V}_{\mathrm{cav}}\mathbf{W_f^\mathsf{T}})^{-1}\mathbf{W_f}\mathbf{m}_{\mathrm{cav}}$$
$$\tag{94}$$

By following the exact procedure as shown above for $q_1(\mathbf{b})$, we can obtain,

$$\log \tilde{Z}_2 + \phi_{\mathrm{cav},2} - \phi_{\mathrm{post}} = \log C_2 - \frac{N}{2}\log(2\pi) - \frac{1}{2}\log|(\Sigma_{\mathbf{y}}/\alpha)| - \frac{1}{2}\mathbf{y}^\mathsf{T}(\Sigma_{\mathbf{y}}/\alpha)^{-1}\mathbf{y} \tag{95}$$

### 3.3  Approximate posterior

Putting the above results together gives the approximate posterior over $\mathbf{b}$ as follows,

$$q_{\mathrm{opt}}(\mathbf{b}) \propto p(\mathbf{b})q_1(\mathbf{b})q_2(\mathbf{b}) \tag{96}$$

$$\propto p(\mathbf{b})\mathcal{N}(\hat{\mathbf{y}}, \mathbf{K_{\hat{f}b}}\mathbf{K_{bb}^{-1}}\mathbf{b}, \Sigma_{\hat{\mathbf{y}}}) \tag{97}$$

$$= \mathcal{N}(\mathbf{a}; \mathbf{K_{bf}}(\mathbf{K_{\hat{f}b}}\mathbf{K_{bb}^{-1}}\mathbf{K_{b\hat{f}}} + \Sigma_{\hat{\mathbf{y}}})^{-1}\hat{\mathbf{y}}, \mathbf{K_{bb}} - \mathbf{K_{bf}}(\mathbf{K_{\hat{f}b}}\mathbf{K_{bb}^{-1}}\mathbf{K_{b\hat{f}}} + \Sigma_{\hat{\mathbf{y}}})^{-1}\mathbf{K_{\hat{f}b}}) \tag{98}$$

where

$$\hat{\mathbf{y}} = \begin{bmatrix} \mathbf{y} \\ \mathbf{y}_a \end{bmatrix} = \begin{bmatrix} \mathbf{y} \\ \mathbf{D_a}\mathbf{S_a^{-1}}\mathbf{m_a} \end{bmatrix}, \quad \mathbf{K_{\hat{f}b}} = \begin{bmatrix} \mathbf{K_{fb}} \\ \mathbf{K_{ab}} \end{bmatrix}, \quad \Sigma_{\hat{\mathbf{y}}} = \begin{bmatrix} \Sigma_{\mathbf{y}} & \mathbf{0} \\ \mathbf{0} & \Sigma_{\mathbf{a}} \end{bmatrix}, \tag{99}$$

and $\Sigma_{\mathbf{y}} = \sigma^2 \mathbf{I} + \alpha \mathrm{diag}\mathbf{Q_f}$, and $\Sigma_{\mathbf{a}} = \mathbf{D_a} + \alpha \mathbf{Q_a}$.

## 3.4 Approximate marginal likelihood

The Power-EP procedure above also provides us an approximation to the marginal likelihood, which can be used to optimise the hyperparameters and the pseudo-inputs,

$$\mathcal{F} = \phi_{\text{post}} - \phi_{\text{prior}} + \frac{1}{\alpha}(\log \tilde{Z}_1 + \phi_{\text{cav},1} - \phi_{\text{post}}) + \frac{1}{\alpha}(\log \tilde{Z}_2 + \phi_{\text{cav},2} - \phi_{\text{post}}) \tag{100}$$

Note that,

$$\Delta_0 = \phi_{\text{post}} - \phi_{\text{prior}} \tag{101}$$

$$= \frac{1}{2} \log |\mathbf{V}| + \frac{1}{2}\mathbf{m}^\intercal \mathbf{V}^{-1}\mathbf{m} - \frac{1}{2} \log |\mathbf{K_{bb}}| \tag{102}$$

$$= -\frac{1}{2} \log |\Sigma_{\hat{\mathbf{y}}}| + \frac{1}{2} \log |\Sigma_a| + \frac{1}{2} \log |\Sigma_{\mathbf{y}}| - \frac{1}{2}\hat{\mathbf{y}}^\intercal \Sigma_{\hat{\mathbf{y}}}^{-1}\hat{\mathbf{y}} + \frac{1}{2}\mathbf{y}^\intercal \Sigma_{\mathbf{y}}^{-1}\mathbf{y} + \frac{1}{2}\mathbf{y_a}^\intercal \Sigma_a^{-1}\mathbf{y_a} \tag{103}$$

$$\Delta_1 = \frac{1}{\alpha}(\log \tilde{Z}_1 + \phi_{\text{cav},1} - \phi_{\text{post}}) \tag{104}$$

$$= \frac{1}{2} \log \frac{|\mathbf{K'_{aa}}|}{|\mathbf{S_a}|} - \frac{1}{2\alpha} \log |\mathbf{I} + \alpha\mathbf{D_a}^{-1}\mathbf{Q_a}| - \frac{1}{2}\mathbf{y_a}^\intercal \Sigma_a^{-1}\mathbf{y_a} + \frac{1}{2}\mathbf{m_a}^\intercal [\mathbf{S_a}^{-1}\mathbf{D_a}\mathbf{S_a}^{-1} - \mathbf{S_a}^{-1}]\mathbf{m_a} \tag{105}$$

$$\Delta_2 = \frac{1}{\alpha}(\log \tilde{Z}_2 + \phi_{\text{cav},2} - \phi_{\text{post}}) \tag{106}$$

$$= -\frac{N}{2} \log(2\pi) + \frac{N(1-\alpha)}{2\alpha} \log(\sigma_y^2) - \frac{1}{2\alpha} \log |\Sigma_{\mathbf{y}}| - \frac{1}{2}\mathbf{y}^\intercal \Sigma_{\mathbf{y}}^{-1}\mathbf{y} \tag{107}$$

Therefore,

$$\mathcal{F} = \log \mathcal{N}(\hat{\mathbf{y}}; 0, \Sigma_{\hat{\mathbf{y}}}) + \frac{N(1-\alpha)}{2\alpha} \log(\sigma_y^2) - \frac{1-\alpha}{2\alpha} \log |\Sigma_{\mathbf{y}}|$$

$$+ \frac{1}{2} \log |\mathbf{K'_{aa}}| - \frac{1}{2} \log |\mathbf{S_a}| + \frac{1}{2} \log |\Sigma_a| - \frac{1}{2\alpha} \log |\mathbf{I} + \alpha\mathbf{D_a}^{-1}\mathbf{Q_a}|$$

$$+ \frac{M_a}{2} \log(2\pi) + \frac{1}{2}\mathbf{m_a}^\intercal [\mathbf{S_a}^{-1}\mathbf{D_a}\mathbf{S_a}^{-1} - \mathbf{S_a}^{-1}]\mathbf{m_a} \tag{108}$$

The limit as $\alpha$ tends to 0 is the variational free energy in eq. (18). This is achieved similar to the batch case as detailed in [6] and by further observing that as $\alpha \to 0$,

$$\frac{1}{2\alpha} \log |\mathbf{I} + \alpha\mathbf{D_a}^{-1}\mathbf{Q_a}| \approx \frac{1}{2\alpha} \log(1 + \alpha\text{tr}(\mathbf{D_a}^{-1}\mathbf{Q_a}) + \mathcal{O}(\alpha^2)) \tag{109}$$

$$\approx \frac{1}{2}\text{tr}(\mathbf{D_a}^{-1}\mathbf{Q_a}) \tag{110}$$

## 3.5 Implementation

In this section, we provide efficient and numerical stable forms for a practical implementation of the above results.

### 3.5.1 The Power-EP approximate marginal likelihood

The first term in eq. (108) can be written as follows,

$$\mathcal{F}_1 = \log \mathcal{N}(\hat{\mathbf{y}}; \mathbf{0}, \mathbf{K_{\hat{f}b}}\mathbf{K_{bb}}^{-1}\mathbf{K_{b\hat{f}}} + \Sigma_{\hat{\mathbf{y}}}) \tag{111}$$

$$= -\frac{N + M_a}{2} \log(2\pi) - \frac{1}{2} \log |\mathbf{K_{\hat{f}b}}\mathbf{K_{bb}}^{-1}\mathbf{K_{b\hat{f}}} + \Sigma_{\hat{\mathbf{y}}}| - \frac{1}{2}\hat{\mathbf{y}}^\intercal (\mathbf{K_{\hat{f}b}}\mathbf{K_{bb}}^{-1}\mathbf{K_{b\hat{f}}} + \Sigma_{\hat{\mathbf{y}}})^{-1}\hat{\mathbf{y}} \tag{112}$$

Let denote $\mathbf{S_y} = \mathbf{K_{\hat{f}b}}\mathbf{K_{bb}}^{-1}\mathbf{K_{b\hat{f}}} + \Sigma_{\hat{\mathbf{y}}}$, $\mathbf{K_{bb}} = \mathbf{L_b}\mathbf{L_b}^\intercal$, $\mathbf{Q_a} = \mathbf{L_q}\mathbf{L_q}^\intercal$, $\mathbf{M_a} = \mathbf{I} + \alpha\mathbf{L_q}^\intercal\mathbf{D_a}^{-1}\mathbf{L_q}$ and $\mathbf{D} = \mathbf{I} + \mathbf{L_b}^{-1}\mathbf{K_{b\hat{f}}}\Sigma_{\hat{\mathbf{y}}}^{-1}\mathbf{K_{\hat{f}b}}\mathbf{L_b}^{-\intercal}$. By using the matrix determinant lemma, we obtain,

$$\log |\mathbf{S_y}| = \log |\mathbf{K_{\hat{f}b}}\mathbf{K_{bb}}^{-1}\mathbf{K_{b\hat{f}}} + \Sigma_{\hat{\mathbf{y}}}| \tag{113}$$

$$= \log |\Sigma_{\hat{\mathbf{y}}}| + \log |\mathbf{I} + \mathbf{L_b}^{-1}\mathbf{K_{b\hat{f}}}\Sigma_{\hat{\mathbf{y}}}^{-1}\mathbf{K_{\hat{f}b}}\mathbf{L_b}^{-\intercal}| \tag{114}$$

$$= \log |\Sigma_{\mathbf{y}}| + \log |\Sigma_{\mathbf{a}}| + \log |D| \tag{115}$$

Note that,

$$\mathbf{K_{b\hat{f}}}\Sigma_{\hat{\mathbf{y}}}^{-1}\mathbf{K_{\hat{f}b}} = \mathbf{K_{bf}}\Sigma_{\mathbf{y}}^{-1}\mathbf{K_{fb}} + \mathbf{K_{ba}}\Sigma_{\mathbf{a}}^{-1}\mathbf{K_{ab}} \tag{116}$$

$$\mathbf{K_{bf}}\Sigma_{\mathbf{y}}^{-1}\mathbf{K_{fb}} = \mathbf{K_{bf}}(\sigma_y^2 I + \alpha \mathbf{Q_f})^{-1}\mathbf{K_{fb}} \tag{117}$$

$$\mathbf{K_{ba}}\Sigma_{\mathbf{a}}^{-1}\mathbf{K_{ab}} = \mathbf{K_{ba}}(\mathbf{D_a} + \alpha \mathbf{Q_a})^{-1}\mathbf{K_{ab}} \tag{118}$$

$$= \mathbf{K_{ba}}(\mathbf{D_a^{-1}} - \alpha \mathbf{D_a^{-1}}\mathbf{L_q}[I + \alpha \mathbf{L_q^\intercal}\mathbf{D_a^{-1}}\mathbf{L_q}]^{-1}\mathbf{L_q^\intercal}\mathbf{D_a^{-1}})\mathbf{K_{ab}} \tag{119}$$

$$= \mathbf{K_{ba}}\mathbf{D_a^{-1}}\mathbf{K_{ab}} - \alpha \mathbf{K_{ba}}\mathbf{D_a^{-1}}\mathbf{L_q}\mathbf{M_a^{-1}}\mathbf{L_q^\intercal}\mathbf{D_a^{-1}}\mathbf{K_{ab}} \tag{120}$$

Using the matrix inversion lemma gives us,

$$\mathbf{S_y^{-1}} = (\mathbf{K_{\hat{f}b}}\mathbf{K_{bb}^{-1}}\mathbf{K_{b\hat{f}}} + \Sigma_{\hat{\mathbf{y}}})^{-1} \tag{121}$$

$$= \Sigma_{\hat{\mathbf{y}}}^{-1} - \Sigma_{\hat{\mathbf{y}}}^{-1}\mathbf{K_{\hat{f}b}}\mathbf{L_b^{-\intercal}}\mathbf{D^{-1}}\mathbf{L_b^{-1}}\mathbf{K_{b\hat{f}}}\Sigma_{\hat{\mathbf{y}}}^{-1} \tag{122}$$

leading to,

$$\hat{\mathbf{y}}^\intercal \mathbf{S_y^{-1}}\hat{\mathbf{y}} = \hat{\mathbf{y}}^\intercal \Sigma_{\hat{\mathbf{y}}}^{-1}\hat{\mathbf{y}} - \hat{\mathbf{y}}^\intercal \Sigma_{\hat{\mathbf{y}}}^{-1}\mathbf{K_{\hat{f}b}}\mathbf{L_b^{-\intercal}}\mathbf{D^{-1}}\mathbf{L_b^{-1}}\mathbf{K_{b\hat{f}}}\Sigma_{\hat{\mathbf{y}}}^{-1}\hat{\mathbf{y}} \tag{123}$$

Note that,

$$\hat{\mathbf{y}}^\intercal \Sigma_{\hat{\mathbf{y}}}^{-1}\hat{\mathbf{y}} = \mathbf{y}^\intercal \Sigma_{\mathbf{y}}^{-1}\mathbf{y} + \mathbf{y_a^\intercal}\Sigma_{\mathbf{y_a}}^{-1}\mathbf{y_a} \tag{124}$$

$$\mathbf{y}^\intercal \Sigma_{\mathbf{y}}^{-1}\mathbf{y} = \mathbf{y}^\intercal (\sigma_y^2 I + \alpha \mathbf{Q_f})^{-1}\mathbf{y} \tag{125}$$

$$\mathbf{y_a^\intercal}\Sigma_{\mathbf{y_a}}^{-1}\mathbf{y_a} = \mathbf{y_a^\intercal}(\mathbf{D_a} + \alpha \mathbf{Q_a})^{-1}\mathbf{y_a} \tag{126}$$

$$= \mathbf{m_a^\intercal}\mathbf{S_a^{-1}}\mathbf{D_a^{-1}}\mathbf{S_a^{-1}}\mathbf{m_a} - \alpha \mathbf{m_a^\intercal}\mathbf{S_a^{-1}}\mathbf{L_q}\mathbf{M_a^{-1}}\mathbf{L_q^\intercal}\mathbf{S_a^{-1}}\mathbf{m_a} \tag{127}$$

$$\text{and} \quad \mathbf{c} = \mathbf{K_{b\hat{f}}}\Sigma_{\hat{\mathbf{y}}}^{-1}\hat{\mathbf{y}} \tag{128}$$

$$= \mathbf{K_{bf}}\Sigma_{\mathbf{y}}^{-1}\mathbf{y} + \mathbf{K_{ba}}\Sigma_{\mathbf{a}}^{-1}\mathbf{y_a} \tag{129}$$

$$= \mathbf{K_{bf}}\Sigma_{\mathbf{y}}^{-1}\mathbf{y} + \mathbf{K_{ba}}\mathbf{S_a^{-1}}\mathbf{m_a} - \alpha \mathbf{K_{ba}}\mathbf{D_a^{-1}}\mathbf{L_q}\mathbf{M_a^{-1}}\mathbf{L_q^\intercal}\mathbf{S_a^{-1}}\mathbf{m_a} \tag{130}$$

Substituting these results back into equation eq. (108),

$$\begin{aligned}
\mathcal{F} = &-\frac{1}{2}\mathbf{y}^\intercal \Sigma_{\mathbf{y}}^{-1}\mathbf{y} + \frac{1}{2}\alpha \mathbf{m_a^\intercal}\mathbf{S_a^{-1}}\mathbf{L_q}\mathbf{M_a^{-1}}\mathbf{L_q^\intercal}\mathbf{S_a^{-1}}\mathbf{m_a} + \frac{1}{2}\mathbf{c}^\intercal \mathbf{L_b^{-\intercal}}\mathbf{D^{-1}}\mathbf{L_b^{-1}}\mathbf{c} \\
&-\frac{1}{2}\log |\mathbf{\Sigma_y}| - \frac{1}{2}\log |\mathbf{D}| - \frac{1}{2}\log |\mathbf{S_a}| + \frac{1}{2}\log |\mathbf{K'_{aa}}| - \frac{1}{2\alpha}\log |\mathbf{M_a}| - \frac{1}{2}\mathbf{m_a^\intercal}\mathbf{S_a^{-1}}\mathbf{m_a} \\
&+\frac{N(1-\alpha)}{2\alpha}\log(\sigma_y^2) - \frac{1-\alpha}{2\alpha}\log |\Sigma_{\mathbf{y}}| - \frac{N}{2}\log(2\pi)
\end{aligned} \tag{131}$$

### 3.5.2 Prediction

We revisit and rewrite the optimal approximate distribution, $q_{\mathrm{opt}}(\mathbf{b})$, using its natural parameters:

$$q_{\mathrm{opt}}(\mathbf{b}) \propto p(\mathbf{b})\mathcal{N}(\hat{\mathbf{y}}, \mathbf{K_{\hat{f}b}}\mathbf{K_{bb}^{-1}}\mathbf{b}, \Sigma_{\hat{\mathbf{y}}}) \tag{132}$$

$$= \mathcal{N}^{-1}(\mathbf{b}; \mathbf{K_{bb}^{-1}}\mathbf{K_{b\hat{f}}}\Sigma_{\hat{\mathbf{y}}}^{-1}\hat{\mathbf{y}}, \mathbf{K_{bb}^{-1}} + \mathbf{K_{bb}^{-1}}\mathbf{K_{b\hat{f}}}\Sigma_{\hat{\mathbf{y}}}^{-1}\mathbf{K_{\hat{f}b}}\mathbf{K_{bb}^{-1}}) \tag{133}$$

The predictive covariance at some test points $\mathbf{s}$ is,

$$\mathbf{V_{ss}} = \mathbf{K_{ss}} - \mathbf{K_{sb}}\mathbf{K_{bb}^{-1}}\mathbf{K_{bs}} + \mathbf{K_{sb}}\mathbf{K_{bb}^{-1}}(\mathbf{K_{bb}^{-1}} + \mathbf{K_{bb}^{-1}}\mathbf{K_{b\hat{f}}}\Sigma_{\hat{\mathbf{y}}}^{-1}\mathbf{K_{\hat{f}b}}\mathbf{K_{bb}^{-1}})^{-1}\mathbf{K_{bb}^{-1}}\mathbf{K_{bs}} \tag{134}$$

$$= \mathbf{K_{ss}} - \mathbf{K_{sb}}\mathbf{K_{bb}^{-1}}\mathbf{K_{bs}} + \mathbf{K_{sb}}\mathbf{L_b^{-\intercal}}(I + \mathbf{L_b^{-1}}\mathbf{K_{b\hat{f}}}\Sigma_{\hat{\mathbf{y}}}^{-1}\mathbf{K_{\hat{f}b}}\mathbf{L_b^{-\intercal}})^{-1}\mathbf{L_b^{-\intercal}}\mathbf{K_{bs}} \tag{135}$$

$$= \mathbf{K_{ss}} - \mathbf{K_{sb}}\mathbf{K_{bb}^{-1}}\mathbf{K_{bs}} + \mathbf{K_{sb}}\mathbf{L_b^{-\intercal}}\mathbf{D^{-1}}\mathbf{L_b^{-\intercal}}\mathbf{K_{bs}} \tag{136}$$

And, the predictive mean,

$$\mathbf{m_s} = \mathbf{K_{sb}}\mathbf{K_{bb}^{-1}}(\mathbf{K_{bb}^{-1}} + \mathbf{K_{bb}^{-1}}\mathbf{K_{b\hat{f}}}\Sigma_{\hat{\mathbf{y}}}^{-1}\mathbf{K_{\hat{f}b}}\mathbf{K_{bb}^{-1}})^{-1}\mathbf{K_{bb}^{-1}}\mathbf{K_{b\hat{f}}}\Sigma_{\hat{\mathbf{y}}}^{-1}\hat{\mathbf{y}} \tag{137}$$

$$= \mathbf{K_{sb}}\mathbf{L_b^{-\intercal}}(I + \mathbf{L_b^{-1}}\mathbf{K_{b\hat{f}}}\Sigma_{\hat{\mathbf{y}}}^{-1}\mathbf{K_{\hat{f}b}}\mathbf{L_b^{-\intercal}})^{-1}\mathbf{L_b^{-1}}\mathbf{K_{b\hat{f}}}\Sigma_{\hat{\mathbf{y}}}^{-1}\hat{\mathbf{y}} \tag{138}$$

$$= \mathbf{K_{sb}}\mathbf{L_b^{-\intercal}}\mathbf{D^{-1}}\mathbf{L_b^{-1}}\mathbf{K_{b\hat{f}}}\Sigma_{\hat{\mathbf{y}}}^{-1}\hat{\mathbf{y}} \tag{139}$$

# 4 Equivalence results

When the hyperparameters and the pseudo-inputs are fixed, $\alpha$-divergence inference for streaming sparse GP regression recovers the batch solutions provided by Power-EP with the same $\alpha$ value. In other words, only a single pass through the data is necessary for Power-EP to converge in sparse GP regression. This result is in a similar vein to the equivalence between sequential inference and batch inference in full GP regression, when the hyperparameters are kept fixed. As an illustrative example, assume that $\mathbf{z}_a = \mathbf{z}_b$ and $\theta$ is kept fixed, and $\{\mathbf{x}_1, \mathbf{y}_1\}$ and $\{\mathbf{x}_2, \mathbf{y}_2\}$ are the first and second data batches respectively. The optimal variational update gives,

$$q_1(\mathbf{a}) \propto p(\mathbf{a}) \exp \int \mathrm{d}\mathbf{f}_1 p(\mathbf{f}_1|\mathbf{a}) \log p(\mathbf{y}_1|\mathbf{f}_1) \tag{140}$$

$$q_2(\mathbf{a}) \propto q_1(\mathbf{a}) \exp \int \mathrm{d}\mathbf{f}_2 p(\mathbf{f}_2|\mathbf{a}) \log p(\mathbf{y}_2|\mathbf{f}_2) \propto p(\mathbf{a}) \exp \int \mathrm{d}\mathbf{f} p(\mathbf{f}|\mathbf{a}) \log p(\mathbf{y}|\mathbf{f}) \tag{141}$$

where $\mathbf{y} = \{\mathbf{y}_1, \mathbf{y}_2\}$ and $\mathbf{f} = \{\mathbf{f}_1, \mathbf{f}_2\}$. Equation (141) is exactly identical to the optimal variational approximation for the batch case of [7], when we group all data batches into one. A similar procedure can be shown for Power-EP. We demonstrate this equivalence in fig. 1.

In addition, in the setting where hyperparameters and the pseudo-inputs are fixed, if pseudo-points are added at each stage at the new data input locations, the method returns the true posterior and marginal likelihood. This equivalence is demonstrated in fig. 2.

Figure 1: Equivalence between the streaming variational approximation and the batch variational approximation when hyperparameters and pseudo-inputs are fixed. The inset numbers are the approximate marginal likelihood (the variational free energy) for each model. Note that the numbers in the batch case are the cumulative sum of the numbers on the left for the streaming case. Small differences, if any, are merely due to numerical computation.

Figure 2: Equivalence between the streaming variational approximation and the exact GP regression when hyperparameters and pseudo-inputs are fixed, and the pseudo-points are at the training points. The inset numbers are the (approximate) marginal likelihood for each model. Note that the numbers in the batch case are the cumulative sum of the numbers on the left for the streaming case. Small differences, if any, are merely due to numerical computation.

## 5 Extra experimental results

### 5.1 Hyperparameter learning on synthetic data

In this experiment, we generated several time series from GPs with known kernel hyperparameters and observation noise. We tracked the hyperparameters as the streaming algorithm learns and plot their traces in figs. 3 and 4. It could be seen that for the smaller lengthscale, we need more pseudo-points to cover the input space and to learn correct hyperparameters. Interestingly, all models including full GP regression on the entire dataset tend to learn bigger noise variances. Overall, the proposed streaming method can track and learn good hyperparameters; and if there is enough pseudo-points, this method performs comparatively to full GP on the entire dataset.

### 5.2 Learning and inference on a toy time series

As shown in the main text, we construct a synthetic time series to demonstrate the learning procedure as data arrives sequentially. Figures 5 and 6 show the results for non-iid and iid streaming settings respectively.

### 5.3 Binary classification

We consider a binary classification task on the benchmark *banana* dataset. In particular, we test two streaming settings, non-iid and iid, as shown in figs. 7 and 8 respectively. In all cases, the streaming algorithm performs well and reaches the performance of the batch case using a sparse variational method [8] (as shown in the right-most plots).

Figure 3: Learnt hyperparameters on a time series dataset, that was generated from a GP with an exponentiated quadratic kernel and with a lengthscale of 0.5. Note the $y-$axis show the difference between the learnt values and the groundtruth.

## 5.4 Sensitivity to the order of the data

We consider the classification task above but now with more (smaller) mini-batches and the order of the batches are varied. The aim is to evaluate the sensitivity of the algorithm to the order of the data. The classification errors as data arrive are included in table 1 and are consistent with what we included in the main text.

Table 1: Classification errors as data arrive in different orders

| Order/Index | 1 | 2 | 3 | 4 | 5 | 6 | 7 | 8 | 9 | 10 |
|---|---|---|---|---|---|---|---|---|---|---|
| Left to Right | 0.255 | 0.145 | 0.1325 | 0.1225 | 0.1075 | 0.11 | 0.105 | 0.1 | 0.0925 | 0.0875 |
| Right to Left | 0.255 | 0.1475 | 0.1325 | 0.12 | 0.105 | 0.1025 | 0.0975 | 0.0925 | 0.09 | 0.095 |
| Random | 0.5025 | 0.2775 | 0.26 | 0.2725 | 0.2875 | 0.1975 | 0.1125 | 0.125 | 0.105 | 0.095 |
| Batch | | | | | | | | | | 0.095 |

## 5.5 Additional plots for the time-series and spatial datasets

In this section, we plot the mean marginal log-likelihood and RMSE against the number of batches for the models in the "speed versus accuracy" experiment in the main text. Fig. 9 shows the results for the time-series datasets while fig. 10 shows the results for the spatial datasets.

## Footnotes

[2]to ensure $q(\mathbf{b})$ is normalised

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

Figure 5: Online regression on a toy time series using variational inference (top) and Power-EP with $\alpha = 0.5$ (bottom), in a non-iid setting. The black crosses are data points (past points are greyed out), the red circles are pseudo-points, and blue lines and shaded areas are the marginal predictive means and confidence intervals at test points.

Figure 6: Online regression on a toy time series using variational inference (top) and Power-EP with $\alpha = 0.5$ (bottom), in an iid setting. The black crosses are data points (past points are greyed out), the red circles are pseudo-points, and blue lines and shaded areas are the marginal predictive means and confidence intervals at test points.

Figure 7: Classifying binary data in a non-iid streaming setting. The right-most plot shows the prediction made by using sparse variational inference on full training data.

Figure 8: Classifying binary data in an iid streaming setting. The right-most plot shows the prediction made by using sparse variational inference on full training data.

Figure 9: Results for time-series datasets with batch sizes 300 and 500. The solid and dashed lines are for $M = 100, 200$ respectively.

Figure 10: Results for spatial data (see fig. 9 for the legend). Solid and dashed lines indicate the results for $M = 400, 600$ pseudo-points respectively.