[Reviews · NeurIPS 2017]

Reviewer 1



This paper presents a variational and alpha-divergence approach specifically designed for the case of Gaussian processes with streaming inputs. The motivation given in the introduction is convincing. The background section is very well presented, sets up the methodology for section 3 and at the same time explains why a batch based approach would be inadequate. Section 3 which is the most important, is however more difficult to follow. It is not a problem to understand how one equation leads to the other, but many important details are left out of the description making it very difficult to understand in a conceptual level. In particular, it'd help to explain the concept behind starting from eq. (5). Further, what is p(a|b) in eq. (6)? It seems to be an important element for our understanding of the method, as it somehow involves the update assumption. I was also wondering whether it'd be possible to arrive at the same bound by starting with the joint distribution and simply applying Jensen's inequality together with (4). That would probably be a clearer presentation. The extension to alpha-divergences and related conversation is interesting, although I would also expect to see a deeper comparison with [9] in the methodological level. This is important since the proposed method converges to [9] for alpha zero, which is what the authors use in practice. The experiments are structured nicely, starting from intuitive demonstration, moving to time-series and spatial data and finishing with classification data. The method does seem to perform well and run fast - it seems to be two time slower than SVI-GP, which still makes it a fast method. However, the experiments do not explore the hypothesis whether the SVI (batch-based) or the streaming based method performs better, simply because they use different approximations (we see that in time zero SVI is already much worse). Since the uncollapsed bound has been used for classification, I'd be keen to see it for the regression tasks too, to be able to compare with SVI-GP. Indeed, the SVI-GP seems to actually reduce the RMSE more than OSGP in Fig. 3. Another useful property to explore / discuss, is what happens when the order of the data is changed. Further, the comparisons do not consider any other competitive method which is tailored to streaming data, such as [9] or [Gijsberts and Metta: Real-time model learning using incremental sparse spectrum GP Regression, 2013]. Finally, it'd be useful to either repeat the plots in Fig 2,3 with number of data in the x-axis (so all methods are aligned) or to simply put those and state the difference in running times. Overall seems like a nice paper, which has good motivation and considers an important problem. However, the key points of the method are not demonstrated or explained adequately. I feel that re-working the presentation and experiments would significantly improve this paper. EDIT after rebuttal period: The authors have clarified most of my concerns and I have raised my score to reflect that.

Reviewer 2



The paper is very well written. The method and underlying motivations are described well. The experiments are clearly described. In particular the movie like figures clearly describe the behaviour of the method on a couple of toy problems.

Reviewer 3



Title: Streaming Sparse Gaussian Process Approximations Summary: The paper proposes a sparse GP approximation to an online setting where data arrives sequentially. The authors provide practical methods for learning and inference in this online setting. Clarity: The paper is well written and clearly introduces the concepts. More detailed comments: 1) The exposition in section 3 is clear, and I also like the fact that the main contribution (3.2) is attached after the mathematical details for a simpler problem. Otherwise the paper would have been harder to read. 2) I am not yet convinced that a sparse method based on pseudo points is the right thing for an online setting: At some point we will run out of pseudo points, and then we will have to make compromises (moving window, ...). Figure 1 points in this direction already. Can you comment on this? 3) l208: Can you give some more details on the statement "all models including full GP regression tend to learn bigger noise variances". It is an observation, but I would like to have an explanation or hypothesis. 4) For the time-series data, are the hyper-parameters learned? 5) Fig. 4: Can you explain a bit the non-monotonic behavior? What is happening? Summary of the review: Overall, a good paper that addresses a problem that is largely ignored by the GP community.